# ESTIMATING FRÉCHET BOUNDS FOR VALIDATING PROGRAMMATIC WEAK SUPERVISION

## ABSTRACT

We develop methods for estimating Fréchet bounds on (possibly high-dimensional) distribution classes in which some variables are continuous-valued. We establish the statistical correctness of the computed bounds under uncertainty in the marginal constraints and demonstrate the usefulness of our algorithms by evaluating the performance of machine learning (ML) models trained with programmatic weak supervision (PWS). PWS is a framework for principled learning from weak supervision inputs (*e.g.* crowdsourced labels, knowledge bases, pre-trained models on related tasks *etc.*), and it has achieved remarkable success in many areas of science and engineering. Unfortunately, it is generally difficult to validate the performance of ML models trained with PWS due to the absence of labeled data. Our algorithms address this issue by estimating sharp lower and upper bounds for performance metrics such as accuracy/recall/precision/F1 score.

## 1 INTRODUCTION

Consider a random vector $(X, Y, Z) \in \mathcal{X} \times \mathcal{Y} \times \mathcal{Z}$ is drawn from an unknown distribution $P$. We assume $\mathcal{X} \subset \mathbb{R}^d$ is arbitrary while $\mathcal{Y}$ and $\mathcal{Z}$ are finite. In this work, we develop and analyze the statistical properties of a method for estimating Fréchet bounds (Rüschendorf, 1991a;b) of the form

$$L \triangleq \inf_{\pi \in \Pi} \mathbb{E}_\pi[g(X, Y, Z)] \quad \text{and} \quad U \triangleq \sup_{\pi \in \Pi} \mathbb{E}_\pi[g(X, Y, Z)] \tag{1.1}$$

when $g$ is a fixed bounded function, with $\Pi$ being the set of distributions $\pi$ for $(X, Y, Z)$ such that the marginal $\pi_{X,Z}$ (resp. $\pi_{Y,Z}$) is identical to the prescribed marginal $P_{X,Z}$ (resp. $P_{Y,Z}$). Our proposed method can efficiently obtain estimates for the bounds by solving convex programs, with the significant advantage that the computational complexity of our algorithm does not scale with the dimensionality of $X$, making it well-suited for applications dealing with high-dimensional data.

In previous work, for example, Fréchet bounds were studied in the financial context (*e.g.*, see Rüschendorf (2017); Bartl et al. (2022)). However, our focus is on applying our methods of estimating Fréchet bounds to the problem of assessing predictors trained using programmatic weak supervision (PWS). PWS is a modern learning paradigm that allows practitioners to train their supervised models without the immediate need for observable labels $Y$ (Ratner et al., 2016; 2017; 2018; 2019; Shin et al., 2021; Zhang et al., 2022). In PWS, practitioners first acquire cheap and abundant weak labels $Z$ through heuristics, crowdsourcing, external APIs, and pretrained models, which serve as proxies for $Y$. Then, the practitioner fits a *label model*, *i.e.*, a graphical model for $P_{Y,Z}$ (Ratner et al., 2016; 2019; Fu et al., 2020; Dong et al., 2022), which, under appropriate modeling assumptions, can be fitted without requiring $Y$'s. Finally, a predictor $h : \mathcal{X} \to \mathcal{Y}$ is trained using a *noise-aware loss* constructed using this fitted label model (Ratner et al., 2016). One major unsolved issue with the weak supervision approach is that even if we knew $P_{Y,Z}$, the risk[1] $R(h) = \mathbb{E}[\ell(h(X), Y)]$ or any other metrics such as accuracy/recall/precision/$F_1$ score are not identifiable (not uniquely specified) because the joint distribution $P_{X,Y}$ is unknown while the marginals $P_{X,Z}$ and $P_{Y,Z}$ are known or can be approximated. As a consequence, any performance metric based on $h$ cannot be estimated without making extra strong assumptions, *e.g.*, $X \perp\!\!\!\perp Y \mid Z$, or assuming the availability of some labeled samples. Unfortunately, these conditions are unlikely to arise in many situations. To overcome this difficulty, we propose a novel method that, among other things, can estimate sharp lower and upper bounds for some metrics of interest. For example, the upper and lower bounds for the accuracy

---

[1] For a given loss function $\ell$ that is not necessarily used during training time.

of a classifier $h$ can be estimated using our method simply by letting $g(x, y, z) = \mathbb{1}[h(x) = y]$ in equation 1.1. At a high level, our method replaces $P_{Y,Z}$ with the fitted label model, and $P_{X,Z}$ with its empirical version in the Fréchet bounds in equation 1.1, and reformulates the problem in terms of a convex optimization. Note that having a good label model is crucial for obtaining accurate bounds for the metric of interest. We summarize our main contributions below:

1. Developing a practical algorithm for estimating the Fréchet bounds in equation 1.1. Our algorithm can be summarized as solving convex programs and is scalable to high-dimensional distributions.

2. Quantifying the uncertainty in the computed bounds due to uncertainty in the prescribed marginals by deriving the asymptotic distribution for our estimators.

3. Applying our method to bounding the accuracy, precision, recall, and $F_1$ score of classifiers trained with weak supervision. This enables practitioners to evaluate classifiers in weak supervision settings *without access to labels*.

## 1.1 RELATED WORK

**Weak supervision:** With the emergence of data-hungry models, the lack of properly labeled datasets has become a major bottleneck in the development of supervised models. One approach to overcome this problem is using programmatic weak supervision (PWS) to train predictors in the absence of high-quality labels $Y$ (Ratner et al., 2016; 2017; 2018; 2019; Shin et al., 2021; Zhang et al., 2022). PWS has shown the potential to solve a variety of tasks in different fields with satisfactory performance. For example, some works have applied weak supervision to named-entity recognition (Lison et al., 2020; Fries et al., 2021; Shah et al., 2023), video frame classification (Fu et al., 2020), bone and breast tumor classification (Varma et al., 2017). More recently, Smith et al. (2022) proposed a new approach to integrating weak supervision and pre-trained large language models (LLMs). Rather than applying LLMs in the usual zero/few-shot fashion, they treat those large models as weak labelers that can be used through prompting to obtain weak signals instead of using hand-crafted heuristics.

A relevant line of research within the realm of weak supervision that is closely related to this work is adversarial learning (Arachie & Huang, 2019; 2021; Mazzetto et al., 2021b;a). Often, adversarial learning aims to learn predictors that perform well in worst-case scenarios. For example, Mazzetto et al. (2021a) develops a method to learn weakly supervised classifiers in the absence of a good label model. In their work, the authors use a small set of labeled data points to constrain the space of possible data distributions and then find a predictor that performs well in the worst-case scenario. Our work relates to this literature in the sense that we are interested in the worst and best-case scenarios over a set of distributions. However, we focus on developing an evaluation method instead of another adversarial learning strategy.

**Set estimation:** It is often the case that the distributions of interest cannot be fully observed, which is generally due to missing or noisy data (Molinari, 2020; Guo et al., 2022). In cases in which practitioners can only observe some aspects of those distributions, *e.g.*, marginal distributions, moments *etc.*, parameters of interest may not be identifiable without strong assumptions due to ambiguity in the observable data. However, set estimation (partial identification) deals with the problem without imposing extra assumptions. This framework allows estimating a set of potential values for the parameters of interest and has been frequently considered in many areas such as microeconometrics (Manski, 1989; 1990; 2003; Molinari, 2020), causal inference (Kallus, 2022; Guo et al., 2022), algorithmic fairness (Fogliato et al., 2020; Prost et al., 2021). Our work is most related to Rüschendorf (1991a;b; 2017); Bartl et al. (2022), which study bounds for the uncertainty of a quantity of interest for a joint distribution that is only partially identified through its marginals, *i.e.*, Fréchet bounds. Compared to the aforementioned works, the novelty of our contribution is proposing a convex optimization algorithm that accurately estimates the Fréchet bounds with proven performance guarantees in a setup that is realized in numerous weak-supervision applications.

## 1.2 NOTATION

We write $\mathbb{E}_Q$ and $\text{Var}_Q$ for the expectation and variance of statistics computed using i.i.d. copies of a random vector $W \sim Q$. Consequently, $\mathbb{P}_Q(A) = \mathbb{E}_Q \mathbb{1}_A$, where $\mathbb{1}_A$ is the indicator of an event $A$. If the distribution is clear by the context, we omit the subscript. If $(a_m)_{m \in \mathbb{N}}$ and $(b_m)_{m \in \mathbb{N}}$ are sequences of scalars, then $a_m = o(b_m)$ is equivalent to $a_m / b_m \to 0$ as $m \to \infty$ and $a_m = b_m + o(1)$ means $a_m - b_m = o(1)$. If $(V^{(m)})_{m \in \mathbb{N}}$ is a sequence of random variables, then (i) $V^{(m)} = o_P(1)$ means that for every $\varepsilon > 0$ we have $\mathbb{P}(|V^{(m)}| > \varepsilon) \to 0$ as $m \to \infty$, (ii) $V^{(m)} = \mathcal{O}_P(1)$ means that for every

$\varepsilon > 0$ there exists a $M > 0$ such that $\sup_{m \in \mathbb{N}} \mathbb{P}(|V^{(m)}| > M) < \varepsilon$, (iii) $V^{(m)} = a_m + o_P(1)$ means $V^{(m)} - a_m = o_P(1)$, (iv) $V^{(m)} = o_P(a_m)$ means $V^{(m)}/a_m = o_P(1)$, and (v) $V^{(m)} = \mathcal{O}_P(a_m)$ means $V^{(m)}/a_m = \mathcal{O}_P(1)$.

## 2 ESTIMATING FRÉCHET BOUNDS

A roadmap to our approach follows: we first reformulate the Fréchet bounds in equation 1.1 into their dual problems, which we discuss in equation 2.1. Then, we replace the non-smooth dual problems with their appropriate smooth approximations, as discussed in equation 2.3. The advantage of having finite $\mathcal{Y}$ and $\mathcal{Z}$ spaces, which is motivated by numerous weak supervision applications, is that we are able to replace the infinite-dimensional primal Fréchet problem in equation 1.1 with a finite-dimensional dual problem. This makes optimization considerably easier.

### 2.1 DUAL FORMULATIONS OF THE BOUNDS AND THEIR APPROXIMATIONS

This section presents a result that allows us to efficiently solve the optimization problems in 1.1 by deriving their dual formulations as finite-dimensional convex programs. Before we dive into the result, let us define a family of matrices denoted by

$$\mathcal{A} \triangleq \left\{ a \in \mathbb{R}^{|\mathcal{Y}| \times |\mathcal{Z}|} \ : \ \textstyle\sum_{y \in \mathcal{Y}} a_{yz} = 0 \text{ for every } z \in \mathcal{Z} \right\}.$$

With this definition in place, we introduce the dual formulation in Theorem 2.1.

**Theorem 2.1.** *Let $g : \mathcal{X} \times \mathcal{Y} \times \mathcal{Z} \to \mathbb{R}$ be a bounded measurable function. Then,*

$$L = \sup_{a \in \mathcal{A}} \mathbb{E}[f_l(X, Z, a)] \quad and \quad U = \inf_{a \in \mathcal{A}} \mathbb{E}[f_u(X, Z, a)], \text{ where} \tag{2.1}$$

$$
\begin{aligned}
f_l(x, z, a) &\triangleq \min_{\bar{y} \in \mathcal{Y}} [g(x, \bar{y}, z) + a_{\bar{y}z}] - \mathbb{E}_{P_{Y|Z}}[a_{Yz} \mid Z = z] \\
f_u(x, z, a) &\triangleq \max_{\bar{y} \in \mathcal{Y}} [g(x, \bar{y}, z) + a_{\bar{y}z}] - \mathbb{E}_{P_{Y|Z}}[a_{Yz} \mid Z = z] .
\end{aligned}
\tag{2.2}
$$

*Moreover, $L$ and $U$ are attained by some optimizers in $\mathcal{A}$.*

Theorem 2.1 remains valid if we maximize/minimize over $\mathbb{R}^{|\mathcal{Y}| \times |\mathcal{Z}|}$ instead of $\mathcal{A}$. However, this is not necessary because the values of $f_l$ and $f_u$ remain identical for the following shifts in $a$: $a_{\cdot z} \leftarrow a_{\cdot z} + b_z$ where $b_z \in \mathbb{R}$. By constraining the set of optimizers to $\mathcal{A}$, we eliminate the possibility of having multiple optimal points.

The computation of these bounds entails finding a minimum or maximum over a discrete set, meaning that straightforward application of their empirical versions could result in optimizing non-smooth functions, which is often challenging. To mitigate this, we consider a smooth approximation of the problem that is found to be useful in handling non-smooth optimization problems (An et al., 2022; Asl & Overton, 2021). We approximate the max and min operators with their "soft" counterparts:

$$
\begin{aligned}
\text{softmin}\{b_1, \cdots, b_K\} &\triangleq -\varepsilon \log[\textstyle\sum_k \exp(-b_k/\varepsilon)] + \varepsilon \log(K) \\
\text{softmax}\{b_1, \cdots, b_K\} &\triangleq \varepsilon \log[\textstyle\sum_k \exp(b_k/\varepsilon)] - \varepsilon \log(K) ,
\end{aligned}
$$

where $\varepsilon > 0$ is a small constant that dictates the level of smoothness. As $\varepsilon$ nears zero, these soft versions of max and min converge to their original non-smooth forms. Using these approximations, we reformulate our dual optimization in equation 2.1 into smooth optimization problems:

$$L_\varepsilon \triangleq \sup_{a \in \mathcal{A}} \mathbb{E}[f_{l,\varepsilon}(X, Z, a)] \quad and \quad U_\varepsilon \triangleq \inf_{a \in \mathcal{A}} \mathbb{E}[f_{u,\varepsilon}(X, Z, a)], \text{ where} \tag{2.3}$$

$$
\begin{aligned}
f_{l,\varepsilon}(x, z, a) &\triangleq -\varepsilon \log\left[ \tfrac{1}{|\mathcal{Y}|} \textstyle\sum_{y \in \mathcal{Y}} \exp\left( \tfrac{g(x,y,z) + a_{yz}}{-\varepsilon} \right) \right] - \mathbb{E}_{P_{Y|Z}}[a_{Yz} \mid Z = z] \\
f_{u,\varepsilon}(x, z, a) &\triangleq \varepsilon \log\left[ \tfrac{1}{|\mathcal{Y}|} \textstyle\sum_{y \in \mathcal{Y}} \exp\left( \tfrac{g(x,y,z) + a_{yz}}{\varepsilon} \right) \right] - \mathbb{E}_{P_{Y|Z}}[a_{Yz} \mid Z = z]
\end{aligned}
\tag{2.4}
$$

and $\varepsilon > 0$ is kept fixed at an appropriate value. As a consequence of Lemma 5 of An et al. (2022), we know that $L_\varepsilon$ and $U_\varepsilon$ are no more than $\varepsilon \log |\mathcal{Y}|$ units from $L$ and $U$. Thus, that distance can be regulated by adjusting $\varepsilon$. For example, if we are comfortable with an approximation error of $10^{-2}$ units when $|\mathcal{Y}| = 2$, we will set $\varepsilon = 10^{-2}/\log(2) \approx .014$.

## 2.2 ESTIMATING THE BOUNDS

In practice, it is not usually possible to solve the optimization problems in 2.3, because we may not have direct access to the distributions $P_{X,Z}$ and $P_{Y|Z}$. We overcome this problem by assuming that we can estimate the distributions using an available dataset.

To this end, let us assume that we have a sample $\{(X_i, Z_i)\}_{i=1}^n \stackrel{\text{iid}}{\sim} P_{X,Z}$, and thus we replace the relevant expectations with $P_{X,Z}$ by its empirical version. Additionally, we have a sequence $\{\hat{P}_{Y|Z}^{(m)}, m \in \mathbb{N}\}$ that estimates $P_{Y|Z}$ with greater precision as $m$ increases. Here, $m$ can be viewed as the size of a sample to estimate $P_{Y|Z}$. Although the exact procedure for estimating the conditional distribution is not relevant to this section, we have discussed in our introductory section that this can be estimated using a *label model* (Ratner et al., 2019; Fu et al., 2020) in applications with weak supervision or in a variety of other ways for applications beyond weak supervision. Later in this section, we will formalize the precision required for the estimates. To simplify our notation, we omit the superscript $m$ in $\hat{P}_{Y|Z}^{(m)}$, whenever it is convenient to do so.

Thus, the Fréchet bounds are estimated as

$$\hat{L}_\varepsilon = \sup_{a \in \mathcal{A}} \frac{1}{n} \sum_{i=1}^n \hat{f}_{l,\varepsilon}(X_i, Z_i, a) \quad \text{and} \quad \hat{U}_\varepsilon = \inf_{a \in \mathcal{A}} \frac{1}{n} \sum_{i=1}^n \hat{f}_{u,\varepsilon}(X_i, Z_i, a), \text{where} \quad (2.5)$$

$$\hat{f}_{l,\varepsilon}(x, z, a) \triangleq -\varepsilon \log \left[ \frac{1}{|\mathcal{Y}|} \sum_{y \in \mathcal{Y}} \exp \left( \frac{g(x,y,z)+a_{yz}}{-\varepsilon} \right) \right] - \mathbb{E}_{\hat{P}_{Y|Z}} [a_{Yz} \mid Z = z]$$

$$\hat{f}_{u,\varepsilon}(x, z, a) \triangleq \varepsilon \log \left[ \frac{1}{|\mathcal{Y}|} \sum_{y \in \mathcal{Y}} \exp \left( \frac{g(x,y,z)+a_{yz}}{\varepsilon} \right) \right] - \mathbb{E}_{\hat{P}_{Y|Z}} [a_{Yz} \mid Z = z] \quad (2.6)$$

In our practical implementations we eliminate the constraint that $\sum_y a_{yz} = 0$ by adding a penalty term $\sum_{z \in \mathcal{Z}} (\sum_{y \in \mathcal{Y}} a_{yz})^2$ to $\hat{U}_\varepsilon$ (and its negative to $\hat{L}_\varepsilon$) and then solve unconstrained convex programs using L-BFGS algorithm (Liu & Nocedal, 1989). Since the penalty term vanishes only when $\sum_y a_{yz} = 0$, we guarantee that the optimal solution is in $\mathcal{A}$.

## 2.3 ASYMPTOTIC PROPERTIES OF THE ESTIMATED BOUNDS

In the following, we state the assumptions required for our asymptotic analysis of $\hat{L}_\varepsilon$ and $\hat{U}_\varepsilon$. Our first assumption is technical, which eliminates certain edge cases and is rather minimal.

**Assumption 2.2.** *For every $y \in \mathcal{Y}$ and $z \in \mathcal{Z}$ the $\mathbb{P}(Y = y \mid Z = z)$ is bounded away from both zero and one, i.e. $\kappa < \mathbb{P}(Y = y \mid Z = z) < 1 - \kappa$ for some $\kappa > 0$.*

In our next assumption, we formalize the degree of precision for the sequence $\{\hat{P}_{Y|Z}^{(m)}, m \in \mathbb{N}\}$ of estimators that we require for desired performances of the bound estimates.

**Assumption 2.3.** *Denote the total variation distance (TV) between probability measures as $d_{\text{TV}}$. For every $z \in \mathcal{Z}$, for some $\lambda > 0$, we have that $d_{\text{TV}}(\hat{P}_{Y|Z=z}^{(m)}, P_{Y|Z=z}) = \mathcal{O}_P(m^{-\lambda})$.*

The asymptotic distributions for the estimated bounds follow.

**Theorem 2.4.** *Assume 2.2 and 2.3, and let $n$ be a function of $m$ such that $n \to \infty$ and $n = o(m^{2\lambda})$ when $m \to \infty$. Then, as $m \to \infty$*

$$\sqrt{n}(\hat{L}_\varepsilon - L_\varepsilon) \Rightarrow N(0, \sigma_{l,\varepsilon}^2) \quad \text{and} \quad \sqrt{n}(\hat{U}_\varepsilon - U_\varepsilon) \Rightarrow N(0, \sigma_{u,\varepsilon}^2)$$

*where $\sigma_{l,\varepsilon}^2 \triangleq \text{Var} f_{l,\varepsilon}(X, Z, a_{l,\varepsilon}^*)$, $\sigma_{u,\varepsilon}^2 \triangleq \text{Var} f_{u,\varepsilon}(X, Z, a_{u,\varepsilon}^*)$, and $a_{l,\varepsilon}^*$ and $a_{u,\varepsilon}^*$ are the unique optimizers to attain $L_\varepsilon$ and $U_\varepsilon$ (equation 2.3).*

The above theorem requires $m^{2\lambda}$ to grow faster than $n$ implying that, through assumption 2.3, $P_{Y|Z}$ is estimated with a precision greater than the approximation error when we replace $P_{X,Z}$ with $\frac{1}{n} \sum_i \delta_{X_i, Z_i}$. This allows us to derive the asymptotic distribution when combined with classical results from M-estimation (see proof in the appendix for more details).

An immediate consequence of the distributional convergence in Theorem 2.4 is that $\hat{L}_\varepsilon$ and $\hat{U}_\varepsilon$ converge in probability to $L_\varepsilon$ and $U_\varepsilon$. Since $L_\varepsilon$ and $U_\varepsilon$ can be arbitrarily close to $L$ and $U$, depending on the choice of $\varepsilon$, so is their estimates, as long as $n$ is large enough.

**Construction of confidence bound:** One interesting use of Theorem 2.4 is that we can construct an approximate confidence interval for the estimates of the bounds. For example, an approximate $1 - \gamma$ confidence interval for $L_\varepsilon$ can is constructed as

$$\hat{I} = \left[ \hat{L}_\varepsilon - \tfrac{\tau_\gamma \hat{\sigma}_{l,\varepsilon}}{\sqrt{n}}, \ \ \hat{L}_\varepsilon + \tfrac{\tau_\gamma \hat{\sigma}_{l,\varepsilon}}{\sqrt{n}} \right], \tag{2.7}$$

where $\tau_\gamma = \Phi^{-1}(1 - \gamma/2)$ and $\hat{\sigma}_{l,\varepsilon}$ is the empirical standard deviation of $f_{l,\varepsilon}(X, Z, \cdot)$, substituting the estimate $\hat{a}$ (solution for the problem in 2.5). For such interval, it holds $\mathbb{P}\left(L_\varepsilon \in \hat{I}\right) \approx 1 - \gamma$, *i.e.*, with approximately $1 - \gamma$ confidence we can say that the true $L_\varepsilon$ is in the interval in equation 2.7. An interval for $U_\varepsilon$ can be constructed similarly.

### 2.4 Extra insights and results

In Section A, we explore three extra theoretical aspects of the Fréchet bounds of interests and our methodology. First, we describe a connection between Fréchet bounds computation and the optimal transport problems (Peyré et al., 2019). Second, we included a theoretical result regarding the size of the difference $U - L$. If this difference is small, then our bounds accurately estimate $\mathbb{E}[g(X, Y, Z)]$ without observing samples from the joint distribution of $(X, Y, Z)$. Intuitively, the new theorem says that if the uncertainty we have about $X$ or $Y$ after observing $Z$ is low, then the bounds $L$ and $U$ should be close (we bound that difference using the concept of conditional entropy). Finally, we have included a theoretical result on label model misspecification. In that result, we study what happens if the label model $\hat{P}_{Y|Z}$ converges to a certain distribution $Q_{Y|Z}$ such that $d_{\mathrm{TV}}(P_{Y|Z=z}, Q_{Y|Z=z}) \leq \delta$ for all $z$. We show that the difference between the resulting bounds from the true bounds is $O(\delta)$.

## 3 Evaluation of model performance in weak supervision

In this section, we describe how to use the ideas presented in Section 2 to estimate non-trivial bounds for the accuracy of a weakly supervised classifier $h$ when no high-quality labels are available. In the standard weak supervision setup, only unlabeled data $(X)$ is available, but the practitioner can extract weak labels $(Z)$ from the available data. More specifically, we assume access to the dataset $\{(X_i, Z_i)\}_{i=1}^m$, i.i.d. with distribution $P_{X,Z}$, used in its entirety to estimate a label model $\hat{P}_{Y|Z}$ (Ratner et al., 2019; Fu et al., 2020) and where part of it, *e.g.*, a random subset of size $n$, is used to estimate bounds. To simplify the exposition, we assume the classifier $h$ is fixed[2]

### 3.1 Risk and accuracy

Let $\ell$ be a generic classification loss function. The risk of a classifier $h$ is defined as $R(h) = \mathbb{E}[\ell(h(X), Y)]$, which cannot be promptly estimated in a weak supervision problem, where we do not observe any $Y$. In this situation, we can make use of our bound estimators in Section 2.2, where we set $g(x, y, z) = \ell(h(x), y)$ to obtain bounds for $R(h)$. Furthermore, we can estimate an uncertainty set for the accuracy of the classification simply by letting $g(x, y, z) = \mathbb{1}[h(x) = y]$.

### 3.2 Precision, recall, and $F_1$ score

For a binary classification problem, where $\mathcal{Y} = \{0, 1\}$, the precision, recall, and $F_1$ score of a classifier $h$ are defined as

$$
\begin{aligned}
p &\triangleq \mathbb{P}(Y = 1 \mid h(X) = 1) = \tfrac{\mathbb{P}(h(X)=1, Y=1)}{\mathbb{P}(h(X)=1)}, \\
r &\triangleq \mathbb{P}(h(X) = 1 \mid Y = 1) = \tfrac{\mathbb{P}(h(X)=1, Y=1)}{\mathbb{P}(Y=1)}, \\
F &\triangleq \tfrac{2}{r^{-1}+p^{-1}} = \tfrac{2\mathbb{P}(h(X)=1, Y=1)}{\mathbb{P}(h(X)=1)+\mathbb{P}(Y=1)}.
\end{aligned}
\tag{3.1}
$$

The quantities $\mathbb{P}(h(X) = 1)$ and $\mathbb{P}(Y = 1)$ in the above definitions are identified, since the marginals $P_{X,Z}$ and $P_{Y,Z}$ are specified in the Fréchet problem in equation 1.1. The $\mathbb{P}(h(X) = 1)$ can be estimated from the full dataset $\{(X_i, Z_i)\}_{i=1}^m$ simply using $\hat{\mathbb{P}}(h(X) = 1) \triangleq \frac{1}{m} \sum_{i=1}^m \mathbb{1}[h(X_i) = 1]$. On the other hand, in most weak supervision applications, $\mathbb{P}(Y = 1)$ is assumed to be known from some prior knowledge or can be estimated from an auxiliary dataset, *e.g.*, using the method described in the appendix of Ratner et al. (2019). Estimating or knowing $\mathbb{P}(Y = 1)$ is required to fit the label model (Ratner et al., 2019; Fu et al., 2020) in the first place, so it is beyond our scope of discussion. Then, we assume we have an accurate estimate $\hat{\mathbb{P}}(Y = 1)$.

---

[2]In practice, it can be obtained using all the data not used to estimate bounds.

The probability $\mathbb{P}(h(X) = 1, Y = 1)$, which is the final ingredient in the definition of precision, recall, and F1 score is not identifiable as $P_{X,Y}$ is unknown. The uncertainty bounds for this quantity can be estimated using our method simply by letting $g(x, y, z) = \mathbb{1}[h(x) = 1 \text{ and } y = 1]$. Let $\hat{L}_\varepsilon$ and $\hat{U}_\varepsilon$ denote the estimated lower and upper bounds for $\mathbb{P}(h(X) = 1, Y = 1)$ obtained using equation 2.5. Naturally, the lower bound estimators for precision, recall, and $F_1$ score are

$$\hat{p}_{l,\varepsilon} \triangleq \frac{\hat{L}_\varepsilon}{\hat{\mathbb{P}}(h(X)=1)}, \ \hat{r}_{l,\varepsilon} \triangleq \frac{\hat{L}_\varepsilon}{\hat{\mathbb{P}}(Y=1)}, \quad \text{and} \quad \hat{F}_{l,\varepsilon} \triangleq \frac{2\hat{L}_\varepsilon}{\hat{\mathbb{P}}(h(X)=1)+\hat{\mathbb{P}}(Y=1)},$$

while the upper bound estimators $\hat{p}_{u,\varepsilon}$, $\hat{r}_{u,\varepsilon}$, and $\hat{F}_{u,\varepsilon}$ are given by substituting $\hat{L}_\varepsilon$ by $\hat{U}_\varepsilon$ above. In the following corollary, we show that the bounds converge asymptotically to normal distributions, which we use for calculating their coverage bounds presented in our applications.

**Corollary 3.1.** *Let $n$ be a function of $m$ such that $n \to \infty$ and $n = o\left(m^{(2\lambda)\wedge 1}\right)$ when $m \to \infty$. Assume the conditions of Theorem 2.4 hold. Then as $m \to \infty$*

$$\sqrt{n}\left(\hat{p}_{l,\varepsilon} - p_{l,\varepsilon}\right) \Rightarrow N(0, \sigma^2_{p,l,\varepsilon}), \ p_{l,\varepsilon} = \frac{L_\varepsilon}{\mathbb{P}(h(X)=1)}, \ \sigma^2_{p,l,\varepsilon} \triangleq \frac{\sigma^2_{l,\varepsilon}}{\mathbb{P}(h(X)=1)^2}$$

$$\sqrt{n}\left(\hat{r}_{l,\varepsilon} - r_{l,\varepsilon}\right) \Rightarrow N(0, \sigma^2_{r,l,\varepsilon}), \ r_{l,\varepsilon} = \frac{L_\varepsilon}{\mathbb{P}(Y=1)}, \ \sigma^2_{r,l,\varepsilon} \triangleq \frac{\sigma^2_{l,\varepsilon}}{\mathbb{P}(Y=1)^2} \tag{3.2}$$

$$\sqrt{n}\left(\hat{F}_{l,\varepsilon} - F_{l,\varepsilon}\right) \Rightarrow N(0, \sigma^2_{F,l,\varepsilon}), \ F_{l,\varepsilon} = \frac{2L_\varepsilon}{\mathbb{P}(h(X)=1)+\mathbb{P}(Y=1)}, \ \sigma^2_{F,l,\varepsilon} \triangleq \frac{4\sigma^2_{l,\varepsilon}}{[\mathbb{P}(h(X)=1)+\mathbb{P}(Y=1)]}$$

*where $L_\varepsilon$, $\sigma^2_{l,\varepsilon}$ are defined in Theorem 2.4. Asymptotic distributions for $\sqrt{n}\left(\hat{p}_{u,\varepsilon} - p_{u,\varepsilon}\right)$, $\sqrt{n}\left(\hat{r}_{u,\varepsilon} - r_{u,\varepsilon}\right)$, and $\sqrt{n}\left(\hat{F}_{u,\varepsilon} - F_{u,\varepsilon}\right)$ are obtained in a similar way by changing $L_\varepsilon$ to $U_\varepsilon$ and $\sigma^2_{l,\varepsilon}$ to $\sigma^2_{u,\varepsilon}$.*

Reiterating our discussion in the final paragraph in Section 2.2, asymptotic distributions are important for constructing confidence intervals for the bounds, which can be done in a similar manner.

## 4 EXPERIMENTS

All experiments are structured to emulate conditions where high-quality labels are inaccessible during training, validation, and testing phases, and all weakly-supervised classifiers are trained using the noise-aware loss (Ratner et al., 2016). To fit the label models, we assume $P_Y$ is known (computed using the full dataset). Unless stated, we use $l_2$-regularized logistic regressors as classifiers, where the regularization strength is determined according to the validation noise-aware loss. In the following, we comment about the used datasets.

**Wrench datasets:** To carry out realistic experiments within the weak supervision setup and study accuracy/F1 score estimation, we utilize datasets incorporated in Wrench (**W**eak Supe**r**vision **Bench**mark) (Zhang et al., 2021). This standardized benchmark platform features real-world datasets and pre-generated weak labels for evaluating weak supervision methodologies. Most of Wrench's datasets are designed for classification tasks, encompassing diverse data types such as tabular, text, and image. For text datasets, we employ the `all-MiniLM-L6-v2` model from the *sentence-transformers*[3] library for feature extraction (Reimers & Gurevych, 2019). Features were extracted for the image datasets before their inclusion in Wrench.

**Hate Speech Dataset (de Gibert et al., 2018):** This dataset contains sentence-level annotations for hate speech in English, sourced from posts from white supremacy forums. It encompasses thousands of sentences classified into either `Hate` (1) or `noHate` (0) categories. This dataset provides an ideal ground for examining recall and precision estimation. Social media moderators aim to maximize the filtering of hate posts, *i.e.*, increasing recall, while ensuring that non-hate content is rarely misclassified as offensive, maintaining high precision. Analogously to the Wrench text datasets, we utilize `all-MiniLM-L6-v2` for feature extraction.

### 4.1 BOUNDING THE PERFORMANCE OF WEAKLY SUPERVISED CLASSIFIERS

In this section, we conduct an empirical study using some of the Wrench and Hate Speech datasets to verify the validity and usefulness of our methodology. We compare results for which $P_{Y|Z}$ is estimated using the true labels $Y$ ("Oracle") and those derived using Snorkel's (Ratner et al., 2017; 2018) default label model with no hyperparameter tuning and a thousand epochs. Such a comparison facilitates an evaluation of our method's efficacy, especially in cases where the label model could be incorrectly specified. Results for other Wrench datasets and one extra label model (FlyingSquid, (Fu et al., 2020)) are presented in Section D.

---

[3]Accessible at https://huggingface.co/sentence-transformers/all-MiniLM-L6-v2.

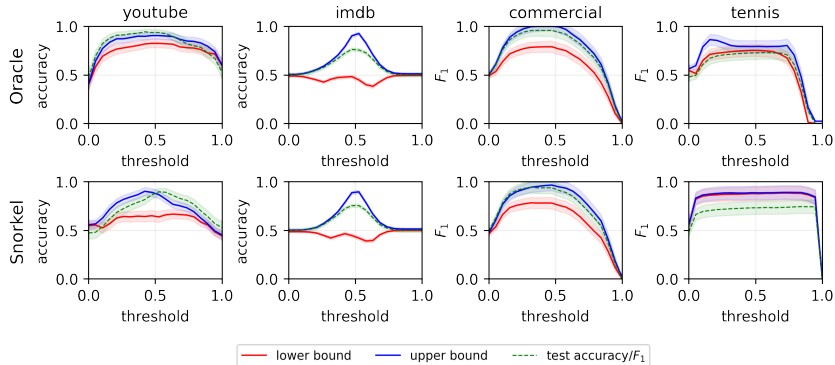

Figure 1: Bounds on classifier accuracies/F1 scores across classification thresholds for the Wrench datasets. Despite potential misspecification in Snorkel's label model, it performs comparably to using labels to estimate $P_{Y|Z}$, giving approximate but meaningful bounds.

In Figure 1, we present bounds on the classifiers' performances across various classification thresholds for binary classification Wrench datasets. A noteworthy observation is that even if Snorkel's label model is potentially misspecified, its performance in estimating bounds mirrors that of when using labels to estimate $P_{Y|Z}$, giving approximate

Table 1: Bounding accuracy in multinomial classification.

| Dataset | Lab. model | Lo. bound | Up. bound | Test acc |
|---------|-----------|-----------|-----------|----------|
| agnews | Oracle | $0.49_{\pm0.01}$ | $0.95_{\pm0.01}$ | $0.82_{\pm0.01}$ |
| | Snorkel | $0.45_{\pm0.01}$ | $0.90_{\pm0.01}$ | $0.80_{\pm0.01}$ |
| chemprot | Oracle | $0.22_{\pm0.02}$ | $0.59_{\pm0.02}$ | $0.41_{\pm0.02}$ |
| | Snorkel | $0.14_{\pm0.02}$ | $0.67_{\pm0.01}$ | $0.39_{\pm0.02}$ |

but meaningful bounds. This suggests that determining bounds remains valuable, irrespective of perfect label model specification. Delving deeper into Figure 1, results for "youtube", "commercial", and "tennis" highlight that our uncertainty about out-of-sample performance is small, even without labeled samples. However, there is a noticeable increase in uncertainty for "imdb", making weakly supervised models deployment riskier without additional validation. Yet, the bounds retain their informative nature. For instance, for those willing to accept the risk, the "imdb" classifier's ideal threshold stands at .5. This is deduced from the flat worst-case and peaking best-case accuracy at this threshold. Table 1 presents some results for "agnews" (4 classes) and "chemprot" (10 classes). The number of possible combinations for values of $Y$ and $Z$ grows exponentially with the number of weak labels, making the "Oracle" estimation of $P_{Y|Z}$ not feasible. Then, we reduce the number of weak labels in "chemprot" from 26 to 10. From Table 1, we can see that both "Oracle" and "Snorkel" approaches produce valid bounds.

Now, we present bounds on the classifiers' precision and recall across different classification thresholds for the hate speech dataset. This dataset did not provide weak labels, so we needed to generate them. We employed four distinct weak labelers. The initial weak labeler functions are based on keywords and terms. Should words or phrases match those identified as hate speech in the lexicon created by Davidson et al. (2017), we categorize the sentence as 1; if not, it's designated 0. The second weak labeler is based on TextBlob's sentiment analyzer (Loria et al., 2018): a negative text polarity results in a 1 classification, while other cases are labeled 0. Our final pair of weak labelers are language models[4], specifically BERT and

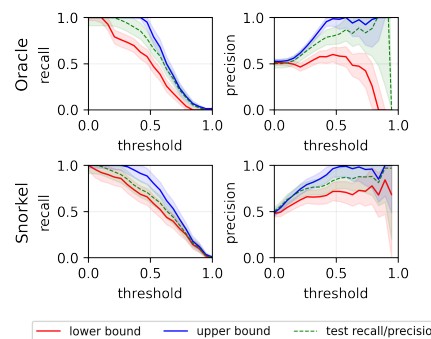

Figure 2: Precision and recall bounds for hate speech detection.

RoBERTa, that have undergone fine-tuning for detecting toxic language or hate speech. Figure 2 presents both recall and precision bounds and test estimates for the weakly-supervised hate speech classifier. Mirroring observations from Figure 1, Snorkel's standard label model gives valuable bounds analogous to scenarios where we employ labels to estimate $P_{Y|Z}$. If used by practitioners,

---

[4]Access the models at `https://huggingface.co/IMSyPP/hate_speech_en` and `https://huggingface.co/s-nlp/roberta_toxicity_classifier`.

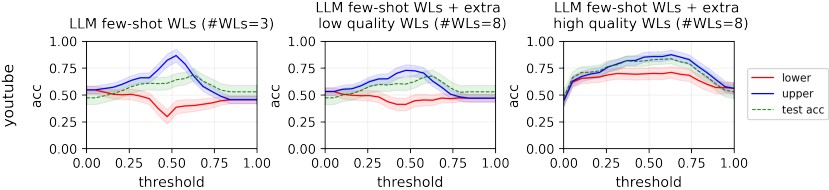

Figure 3: Performance bounds for classifiers on the YouTube dataset, initially relying solely on few-shot weak labels obtained via prompts to the LLM `Llama-2-13b-chat-hf`. The progression of plots illustrates the comparative impact of integrating "high-quality" labels from Wrench versus synthetically generated "low-quality" labels. Evidently, the addition of "high-quality" labels significantly enhances the bounds, underscoring their superior utility over "low-quality" labels for optimal classification of SPAM and HAM comments.

Figure 2 could help trade-off recall and precision by choosing an appropriate classification threshold in the absence of high-quality labels.

## 4.2 CHOOSING A SET OF WEAK LABELS

In this experiment, our aim is to examine how our proposed approach performs under the influence of highly informative weak labels as opposed to scenarios with less informative weak labels. Using the YouTube dataset provided by Wrench, we attempt to classify YouTube comments into categories of SPAM or HAM, leveraging Snorkel to estimate $P_{Y|Z}$. Inspired by Smith et al. (2022), we craft three few-shot weak labelers by prompting[5] the large language model (LLM) called `Llama-2-13b-chat-hf`[6]. For each dataset entry, we pose three distinct queries to the LLM. Initially, we inquire if the comment is SPAM or HAM. Next, we provide clear definitions of SPAM and HAM, then seek the classification from LLM. In the third prompt, leveraging in-context learning ideas (Dong et al., 2022), we provide five representative comments labeled as SPAM/HAM prior to requesting the LLM's verdict on the comment in question. In cases where LLM's response diverges from SPAM or HAM, we interpret it as LLM's abstention.

After obtaining this triad of weak labels, we analyze two situations. Initially, we integrate the top five[7] weak labels ("high-quality" labels) from Wrench. In the subsequent scenario, we synthetically generate weak labels ("low-quality" labels) that bear no linkage with $Y$. The first plot in Figure 3 depicts the bounds of our classifier based solely on weak few-shot labels, which unfortunately do not provide substantial insights. Enhancing the bounds requires the inclusion of additional weak labels. Yet, as indicated by the subsequent pair of plots, it becomes evident that only the incorporation of "high-quality" weak labels results in significant shrinkage and upward shift of the bounds. As confirmed by the test accuracy, if a practitioner had used our method to select the set of weak labels, that would have led to a significant boost in performance.

### 4.2.1 MODEL SELECTION STRATEGIES USING THE FRÉCHET BOUNDS

In Sections 4.1 and 4.2, we implicitly touched on the topic of model selection when discussing threshold and weak label selection. Here, we explicitly discuss the use of our Fréchet bounds for model selection purposes. Consider a set of possible models $\mathcal{H} \triangleq \{h_1, \cdots, h_K\}$ from which we wish to find the best model according to a specific metric, *e.g.*, accuracy, or F1 score. We consider three approaches for model selection using the Fréchet bounds: choosing the model with the best possible (i) lower bound, (ii) upper bound, and (iii) average of lower and upper bounds on the metric of interest. Strategy (i) works well for the worst-case scenario and can be seen as the distributionally robust optimization (DRO) (Chen et al., 2020) solution when the uncertainty set is given by $\Pi$ in equation 1.1, while (ii) is suitable for an optimistic scenario, and (iii) is suggested when one wants to balance between the worst- and best-case scenarios. Please check Section A.4 for more details.

We conduct an experiment to test the idea of using Fréchet bounds for model selection purposes. In this experiment, we select multilayer-perceptrons (MLPs). The considered MLPs have one hidden layer with a possible number of neurons in $\{50, 100\}$. Training is carried out with Adam (Kingma & Ba, 2014), with possible learning rates in $\{.1, .001\}$ and weight decay ($l_2$ regularization parameter) in $\{.1, .001\}$. For those datasets that use the F1 score as the evaluation metric, we also tune the classification threshold in $\{.2, .4, .5, .6, .8\}$ (otherwise, they return the most probable class as a

---

[5]More details regarding the prompts can be found in the appendix.

[6]Available at https://huggingface.co/meta-llama/Llama-2-13b-chat-hf.

[7]The most informative weak labels are determined based on their alignment with the true label.

prediction). In total, $\mathcal{H}$ is composed of $8$ trained models when evaluating accuracy and $40$ models when evaluating the F1 score. We also consider directly using the label model (Snorkel (Ratner et al., 2018)) to select models. For example, when the metric considered is accuracy, *i.e.*, we use select the model $\arg\max_{h_k \in \mathcal{H}} \frac{1}{n} \sum_{i=1}^{n} \mathbb{E}_{\hat{P}_{Y|Z}} \mathbb{1}[h_k(X) = Y \mid Z = Z_i]$, which is a natural choice when $X \perp\!\!\!\perp Y \mid Z$. As benchmarks, we consider having a few labeled samples.

In Table 3, we report a subset of our results (please check Section A.4 for the full set of results). In this table, we report the average test scores of the chosen models over 5 repetitions for different random seeds (standard deviation report as subscript). We can

Table 2: Performance of selected models

| | Lower bound | Upper bound | Bounds avg | Label model | Labeled ($n = 100$) |
|---|---|---|---|---|---|
| agnews | $0.80_{\pm 0.00}$ | $0.80_{\pm 0.00}$ | $0.80_{\pm 0.00}$ | $0.80_{\pm 0.00}$ | $0.80_{\pm 0.00}$ |
| imdb | $0.50_{\pm 0.00}$ | $0.69_{\pm 0.00}$ | $0.74_{\pm 0.01}$ | $0.75_{\pm 0.00}$ | $0.75_{\pm 0.00}$ |
| tennis | $0.77_{\pm 0.00}$ | $0.77_{\pm 0.00}$ | $0.77_{\pm 0.00}$ | $0.77_{\pm 0.00}$ | $0.70_{\pm 0.11}$ |
| sms | $0.23_{\pm 0.01}$ | $0.66_{\pm 0.01}$ | $0.66_{\pm 0.01}$ | $0.23_{\pm 0.01}$ | $0.73_{\pm 0.03}$ |
| commercial | $0.91_{\pm 0.00}$ | $0.78_{\pm 0.00}$ | $0.91_{\pm 0.00}$ | $0.91_{\pm 0.00}$ | $0.85_{\pm 0.04}$ |

extract some lessons from the table. First, using metrics derived from the Fréchet bounds is most useful when our uncertainty about the model performance is low, *e.g.*, "'commercial" and "tennis" in Figure 1. In those cases, using our metrics for model selection gives better results even when compared to a labeled validation set of size $n = 100$. Moreover, once the practitioner knows that the uncertainty is low, using the label model approach will also lead to good results. Second, in some cases using the bounds average can be more beneficial than using the label model approach, which implicitly assumes $X \perp\!\!\!\perp Y \mid Z$. However, using the lower bound might be good if the practitioner wants to be robust/safe even though those results typically do not stand out in Wrench datasets.

## 5 CONCLUSION

**Limitations.** We highlight two main limitations of our work. Firstly, our method and theoretical results can only be applied to cases in which $\mathcal{Y}$ and $\mathcal{Z}$ are finite sets, *e.g.*, in classification problems. In the case of general $\mathcal{Y}$ and $\mathcal{Z}$, the dual formulation in Theorem 2.1 can probably be extended without too much effort. However, the optimization problem we would have to solve in that case requires optimizing lower/upper bounds over spaces of functions which can prove to be challenging both computationally and theoretically. The second limitation regards the convergence of the label model to $\hat{P}_{Y|Z}$ to the true conditional distribution $P_{Y|Z}$. One issue that needs to be considered is the label model specification, *e.g.*, through specifying a dependency graph. If the label model is not correctly specified, it should not converge to the true target distribution; thus, estimated bounds might not be meaningful. Another issue is that if $|\mathcal{Z}|$ is large, the label model's convergence might be slow, and a big unlabeled dataset is needed for the bounds to be accurately estimated. Using a more curated but smaller set of weak labels may be more advantageous for bounds estimation, and using the ideas presented in Section 4.2 can be useful in that case.

**Extensions.** One possible extension of our method is the accommodation of more general $\mathcal{Y}$ and $\mathcal{Z}$ to permit bound estimation to more general tasks such as regression. Another possible extension is the application of the ideas presented in this work in different fields of machine learning or statistics. One could consider applying our ideas to the problem of "statistical matching" (SM) (D'Orazio et al., 2006; Conti et al., 2016; D'Orazio, 2019; Lewaa et al., 2021; Conti et al., 2019), for example. The classic formulation of SM involves observing two distinct datasets that contain replications of $(X, Z)$ and $(Y, Z)$, but the triplet $(X, Y, Z)$ is never observed. The primary goal is to make inferences about the relationship between $X$ and $Y$. For example, one extensively studied instance involves matching data from the Programme for International Student Assessment (PISA) and the Teaching and Learning International Survey (TALIS) (Kaplan & Turner, 2012; Kaplan & McCarty, 2013; Leunda Iztueta et al., 2017). PISA provides data about students and their schools across various countries, while TALIS furnishes data about teachers and their schools. In this context, for a given country, the matching variables $Z$ are school attributes like size, with $X$ representing the average student grades within a school and $Y$ denoting the teachers' motivation, for instance. For instance, if our focus is on bounding $\mathbb{P}((X, Y) \in B) = \mathbb{E}[\mathbb{1}_B(X, Y)]$ for a certain event $B$, we could define $g(x, y, z) = \mathbb{1}_B(x, y)$ and apply our method.

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

## A  EXTRA CONTENT

### A.1  CONNECTION TO OPTIMAL TRANSPORT

The optimizations in the Frechet bounds equation 1.1 can be connected to an optimization problem. We only explain this connection for the lower bound, but the connection to the upper bound is quite similar. The optimization lower bound is

$$\inf_{\substack{\pi_{X,Z}=P_{X,Z}\\ \pi_{Y|Z}=P_{Y|Z}}} \mathbb{E}_\pi[g(X,Y,Z)] =$$

$$= \inf_{\substack{\pi_{X,Z}=P_{X,Z}\\ \pi_{Y|Z}=P_{Y|Z}}} \sum_z \mathbb{P}(Z=z)\mathbb{E}_\pi[g(X,Y,Z) \mid Z=z]$$

$$= \sum_z \mathbb{P}(Z=z)\left\{\inf_{\substack{\pi_{X|Z=z}=P_{X|Z=z}\\ \pi_{Y|Z=z}=P_{Y|Z=z}}} \mathbb{E}_{\pi_{X,Y|Z=z}}[g(X,Y,z)]\right\}$$

where we notice that the inner minimization is an optimal transport problem between the probability distributions $P_{X|Z=z}$ and $P_{Y|Z=z}$ with the cost function $d_z(x,y)=g(x,y,z)$.

### A.2  WHEN ARE THE BOUNDS INFORMATIVE?

We say that the bounds $L$ and $U$ are informative if the difference $U-L$ is small, *i.e.*, we can estimate $\mathbb{E}[g(X,Y,Z)]$ with high precision even if the joint random vector $(X,Y,Z)$ is never observed. Theorem A.1 gives an upper bound for $U-L$ based on the marginal distributions $P_{X,Z}$ and $P_{Y,Z}$.

Intuitively, Theorem A.1 says that if the uncertainty we have about $X$ or $Y$ after observing $Z$ is low, then the bounds $L$ and $U$ should be close. For example, take the extreme case where $Y=Z$: knowing $Z$ tells us exactly the value of $Y$, and we could perfectly label every data point $(X,Z)$, recovering the triplet $(X,Y,Z)$. In that case, $H(Y \mid Z)=0$ and, consequently, $U=L$.

**Theorem A.1.** *Let $L$ and $U$ be defined as in equation equation 1.1. Then*

$$U - L \le \sqrt{8\|g\|_\infty^2 \min\left(H(X \mid Z), H(Y \mid Z)\right)}$$

*where $H(X \mid Z)$ (resp. $H(Y \mid Z)$) denotes the conditional entropy of $X$ (resp. $Y$) given $Z$.*

*Proof.* In this proof, we shall assume $X$ is a continuous random variable for clarity, even though this is not strictly needed.

Let $Q_{X,Y,Z}^{(1)}$ and $Q_{X,Y,Z}^{(2)}$ be two distributions in $\Pi$, *i.e.*, with marginals $P_{X,Z}$ and $P_{Y,Z}$. Let the densities of $Q_{X,Y,Z}^{(1)}$ and $Q_{X,Y,Z}^{(2)}$ as $q_{X,Y,Z}^{(1)}$ and $q_{X,Y,Z}^{(2)}$. Then, see that

$$\left|\int g\mathrm{d}Q_{X,Y,Z}^{(1)} - \int g\mathrm{d}Q_{X,Y,Z}^{(2)}\right| =$$

$$= \left|\int \sum_{y,z} g(x,y,z)(q_{X,Y,Z}^{(1)}(x,y,z) - q_{X,Y,Z}^{(2)}(x,y,z))\mathrm{d}x\right|$$

$$\le \int \sum_{y,z} |g(x,y,z)| \cdot |q_{X,Y,Z}^{(1)}(x,y,z) - q_{X,Y,Z}^{(2)}(x,y,z)|\mathrm{d}x$$

$$\le 2\|g\|_\infty \sum_z p_Z(z)\frac{1}{2}\int \sum_y |q_{X,Y|Z}^{(1)}(x,y \mid z) - q_{X,Y|Z}^{(2)}(x,y \mid z)|\mathrm{d}x$$

$$= 2\|g\|_\infty \mathbb{E}\left[d_{\mathrm{TV}}(Q_{X,Y|Z}^{(1)}, Q_{X,Y|Z}^{(2)})\right]$$

$$\le 2\|g\|_\infty \left\{\mathbb{E}\left[d_{\mathrm{TV}}(Q_{X,Y|Z}^{(1)}, P_{X|Z}P_{Y|Z})\right] + \mathbb{E}\left[d_{\mathrm{TV}}(Q_{X,Y|Z}^{(2)}, P_{X|Z}P_{Y|Z})\right]\right\} \tag{A.1}$$

$$\le 2\|g\|_\infty \left\{\mathbb{E}\left[\sqrt{\tfrac{1}{2}\mathrm{KL}(Q_{X,Y|Z}^{(1)}||P_{X|Z}P_{Y|Z})}\right] + \mathbb{E}\left[\sqrt{\tfrac{1}{2}\mathrm{KL}(Q_{X,Y|Z}^{(2)}||P_{X|Z}P_{Y|Z})}\right]\right\} \tag{A.2}$$

$$\le 2\|g\|_\infty \left\{\sqrt{\tfrac{1}{2}\mathbb{E}\left[\mathrm{KL}(Q_{X,Y|Z}^{(1)}||P_{X|Z}P_{Y|Z})\right]} + \sqrt{\tfrac{1}{2}\mathbb{E}\left[\mathrm{KL}(Q_{X,Y|Z}^{(2)}||P_{X|Z}P_{Y|Z})\right]}\right\} \tag{A.3}$$

$$\le 2\|g\|_\infty \left\{\sqrt{\tfrac{1}{2}H(X \mid Z)} + \sqrt{\tfrac{1}{2}H(X \mid Z)}\right\} \tag{A.4}$$

$$\le \sqrt{8\|g\|_\infty^2 H(X \mid Z)}$$

where step A.1 is justified by triangle inequality, step A.2 is justified by Pinsker's inequality, step A.3 is justified by Jensen's inequality, and step A.4 is justified by the fact that both expected KL terms are conditional mutual information terms, which can be bounded by the conditional entropy. Following the same idea, we can show that $\left| \int g \mathrm{d} Q_{X,Y,Z}^{(1)} - \int g \mathrm{d} Q_{X,Y,Z}^{(2)} \right| \leq \sqrt{8 \|g\|_\infty^2 H(Y \mid Z)}$. Consequently,

$$
\begin{aligned}
\left| \int g \mathrm{d} Q_{X,Y,Z}^{(1)} - \int g \mathrm{d} Q_{X,Y,Z}^{(2)} \right| &\leq \min \left( \sqrt{8 \|g\|_\infty^2 H(X \mid Z)}, \sqrt{8 \|g\|_\infty^2 H(Y \mid Z)} \right) \\
&= \sqrt{8 \|g\|_\infty^2 \min \left( H(X \mid Z), H(Y \mid Z) \right)}.
\end{aligned}
$$

Because this statement is valid for any distributions $Q_{X,Y,Z}^{(1)}$ and $Q_{X,Y,Z}^{(2)}$ in $\Pi$, we have that

$$
U - L \leq \sqrt{8 \|g\|_\infty^2 \min \left( H(X \mid Z), H(Y \mid Z) \right)}
$$

$\square$

### A.3 LABEL MODEL MISSPECIFICATION

In this section, we evaluate the performance of Fréchet bounds when the label model $P_{Y|Z}$ is misspecified. Assume that

**Assumption A.2.** *For some $\delta > 0$ and every $z \in \mathcal{Z}$ we have that*

$$
d_{\mathrm{TV}} \left( Q_{Y|Z=z}, P_{Y|Z=z} \right) \leq \delta. \tag{A.5}
$$

Then we show that

**Theorem A.3.** *Recall from equation equation 2.3 that $L_\epsilon$ and $U_\epsilon$ are the smoothened upper and lower Fréchet bounds with the true $P_{Y|Z=z}$. Additionally, define $\check{L}_\epsilon$ and $\check{U}_\epsilon$ as the same Fréchet bounds, but with misspecified $Q_{Y|Z=z}$, i.e.*

$$
\check{L}_\epsilon \triangleq \sup_{a \in \mathcal{A}} \mathbb{E}[\check{f}_{l,\varepsilon}(X, Z, a)] \quad and \quad \check{U}_\epsilon \triangleq \inf_{a \in \mathcal{A}} \mathbb{E}[\check{f}_{u,\varepsilon}(X, Z, a)], \tag{A.6}
$$

*where*

$$
\begin{aligned}
\check{f}_{l,\varepsilon}(x, z, a) &\triangleq -\varepsilon \log \left[ \frac{1}{|\mathcal{Y}|} \sum_{y \in \mathcal{Y}} \exp \left( \frac{g(x,y,z) + a_{yz}}{-\varepsilon} \right) \right] - \mathbb{E}_{Q_{Y|Z}}[a_{Yz} \mid Z = z] \\
\check{f}_{u,\varepsilon}(x, z, a) &\triangleq \varepsilon \log \left[ \frac{1}{|\mathcal{Y}|} \sum_{y \in \mathcal{Y}} \exp \left( \frac{g(x,y,z) + a_{yz}}{\varepsilon} \right) \right] - \mathbb{E}_{Q_{Y|Z}}[a_{Yz} \mid Z = z]
\end{aligned} \tag{A.7}
$$

*If $Q_{Y|Z}$ satisfies the assumption 2.2, then for some $C > 0$ which is independent of $\delta > 0$ we have*

$$
|\check{L}_\epsilon - L_\epsilon| \vee |\check{U}_\epsilon - U_\epsilon| \leq C\delta. \tag{A.8}
$$

*Proof of Theorem A.3.* We only prove the theorem for upper Fréchet bound. The proof of lower bound is similar.

First, notice that

$$
\begin{aligned}
\check{f}_{u,\varepsilon}(x, z, a) - f_{u,\epsilon}(x, z, a) &= \\
&= \varepsilon \log \left[ \frac{1}{|\mathcal{Y}|} \sum_{y \in \mathcal{Y}} \exp \left( \frac{g(x,y,z) + a_{yz}}{\varepsilon} \right) \right] - \mathbb{E}_{Q_{Y|Z}}[a_{Yz} \mid Z = z] \\
&\quad - \varepsilon \log \left[ \frac{1}{|\mathcal{Y}|} \sum_{y \in \mathcal{Y}} \exp \left( \frac{g(x,y,z) + a_{yz}}{\varepsilon} \right) \right] + \mathbb{E}_{P_{Y|Z}}[a_{Yz} \mid Z = z] \\
&= \mathbb{E}_{P_{Y|Z}}[a_{Yz} \mid Z = z] - \mathbb{E}_{Q_{Y|Z}}[a_{Yz} \mid Z = z]
\end{aligned}
$$

and thus

$$
|\check{f}_{u,\varepsilon}(x, z, a) - f_{u,\epsilon}(x, z, a)| \tag{A.9}
$$
$$
= \left| \mathbb{E}_{P_{Y|Z}}[a_{Yz} \mid Z = z] - \mathbb{E}_{Q_{Y|Z}}[a_{Yz} \mid Z = z] \right| \tag{A.10}
$$
$$
\leq \|a\|_\infty \times 2 d_{\mathrm{TV}} \left( Q_{Y|Z=z}, P_{Y|Z=z} \right) \quad \text{(using } a^\top b \leq \|a\|_\infty \|b\|_1 \text{)} \tag{A.11}
$$
$$
\leq 2\|a\|_\infty \delta. \quad \text{(Assumption A.2)} \tag{A.12}
$$

Defining

$$a^\star \triangleq \arg\min_a \mathbb{E}[\check{f}_{u,\varepsilon}(X, Z, a)], \quad \check{a} \triangleq \arg\min_a \mathbb{E}[\check{f}_{u,\varepsilon}(X, Z, a)], \tag{A.13}$$

we now establish that

$$|\check{U}_\epsilon - U_\epsilon| \le 2(\|a^\star\|_\infty \vee \|\check{a}\|_\infty)\delta. \tag{A.14}$$

This is easily established from the following arguments:

$$
\begin{aligned}
\check{U}_\epsilon &= \mathbb{E}[\check{f}_{u,\varepsilon}(X, Z, \check{a})] \\
&\le \mathbb{E}[\check{f}_{u,\varepsilon}(X, Z, a^\star)] && (\check{a} \text{ is the minimizer}) \\
&\le \mathbb{E}[f_{u,\varepsilon}(X, Z, a^\star)] + 2\|a^\star\|_\infty\delta && (\text{using eq. equation A.12}) \\
&= U_\epsilon + 2\|a^\star\|_\infty\delta
\end{aligned}
\tag{A.15}
$$

and similarly

$$U_\epsilon = \mathbb{E}[f_{u,\varepsilon}(X, Z, a^\star)] \le \mathbb{E}[f_{u,\varepsilon}(X, Z, \check{a})] \le \mathbb{E}[\check{f}_{u,\varepsilon}(X, Z, \check{a})] + 2\|\check{a}\|_\infty\delta = \check{U}_\epsilon + 2\|a^\star\|_\infty\delta. \tag{A.16}$$

Notice from Lemma C.5 that $a^\star$ is bounded. Since $Q_{Y|Z}$ satisfies assumption 2.2, one can make a similar argument for $\check{a}$ that it is also bounded. Thus, we can find a $C > 0$, which is independent of $\delta > 0$, such that

$$2(\|a^\star\|_\infty \vee \|\check{a}\|_\infty) \le C$$

and the theorem holds for this $C$. □

## A.4 MODEL SELECTION USING PERFORMANCE BOUNDS

In this section, we propose and empirically evaluate three strategies for model selection using our estimated bounds in Equation 2.5.

### A.4.1 INTRODUCING MODEL SELECTION STRATEGIES

Assume, for example, $g(x, y, z) = \mathbb{1}[h(x) = y]$ for a given classifier $h$, *i.e.*, we conduct model selection based on accuracy, even though we can easily extend the same idea to different choices of metrics, such as F1 score. The model selection problem consists of choosing the best model from a set $\mathcal{H} \triangleq \{h_1, \cdots, h_K\}$ in order to maximize out-of-sample accuracy. Define $\hat{L}_\varepsilon(h)$ and $\hat{U}_\varepsilon(h)$ as the estimated accuracy lower and upper bounds for a certain model $h$. The first strategy is to choose the model with *highest* accuracy lower bound, *i.e.*,

$$h^*_{\text{lower}} = \arg\max_{h_k \in \mathcal{H}} \hat{L}_\varepsilon(h_k)$$

Maximizing the accuracy lower bound approximates the distributionally robust optimization (DRO) (Chen et al., 2020) solution when the uncertainty set is given by $\Pi$ in 1.1. That is, we optimize for the worst-case distribution in the uncertainty set. Analogously, we can choose the model that optimizes the best-case scenario

$$h^*_{\text{upper}} = \arg\max_{h_k \in \mathcal{H}} \hat{U}_\varepsilon(h_k).$$

Moreover, if we want to guarantee that both worst and best-case scenarios are not bad, we can optimize the average of upper and lower bounds, *i.e.*,

$$h^*_{\text{avg}} = \arg\max_{h_k \in \mathcal{H}} \frac{\hat{L}_\varepsilon(h_k) + \hat{U}_\varepsilon(h_k)}{2}.$$

### A.4.2 EXPERIMENT SETUP

In this experiment, we select multilayer-perceptrons (MLPs). The considered MLPs have one hidden layer with a possible number of neurons in $\{50, 100\}$. Training is carried out with Adam (Kingma & Ba, 2014), with possible learning rates in $\{.1, .001\}$ and weight decay ($l_2$ regularization parameter) in $\{.1, .001\}$. For those datasets that use the F1 score as the evaluation metric, we also tune the classification threshold in $\{.2, .4, .5, .6, .8\}$ (otherwise, they return the most probable class as a prediction). In total, $\mathcal{H}$ is composed of 8 trained models when evaluating accuracy and 40 models

when evaluating the F1 score. We also consider directly using the label model (Snorkel (Ratner et al., 2018)) to select models. For example, when the metric considered is accuracy, *i.e.*, we use

$$h^*_{\text{label\_model}} = \arg\max_{h_k \in \mathcal{H}} \frac{1}{n} \sum_{i=1}^{n} \mathbb{E}_{\hat{P}_{Y|Z}} \mathbb{1}[h_k(X) = Y \mid Z = Z_i],$$

which is a natural choice when $X \perp\!\!\!\perp Y \mid Z$. As benchmarks, we consider having a few labeled samples.

In Table 3, we report the average test scores of the chosen models over 5 repetitions for different random seeds (standard deviation report as subscript). The main message here is that, for those datasets in which our uncertainty about the score (given by the different upper and lower bounds) is small, *e.g.*, "'commercial" and "tennis", using our approaches leads to much better results when compared to using small sample sizes.

Table 3: Performance of selected models

| | Lower bound | Upper bound | Bounds avg | Label model | Labeled ($n = 10$) | Labeled ($n = 25$) | Labeled ($n = 50$) | Labeled ($n = 100$) |
|---|---|---|---|---|---|---|---|---|
| agnews | $0.80_{\pm 0.00}$ | $0.80_{\pm 0.00}$ | $0.80_{\pm 0.00}$ | $0.80_{\pm 0.00}$ | $0.80_{\pm 0.00}$ | $0.80_{\pm 0.00}$ | $0.80_{\pm 0.00}$ | $0.80_{\pm 0.00}$ |
| trec | $0.49_{\pm 0.01}$ | $0.50_{\pm 0.01}$ | $0.50_{\pm 0.01}$ | $0.50_{\pm 0.02}$ | $0.45_{\pm 0.09}$ | $0.50_{\pm 0.01}$ | $0.50_{\pm 0.01}$ | $0.49_{\pm 0.02}$ |
| semeval | $0.55_{\pm 0.01}$ | $0.53_{\pm 0.00}$ | $0.55_{\pm 0.01}$ | $0.55_{\pm 0.01}$ | $0.50_{\pm 0.05}$ | $0.52_{\pm 0.04}$ | $0.54_{\pm 0.01}$ | $0.54_{\pm 0.01}$ |
| chemprot | $0.48_{\pm 0.00}$ | $0.50_{\pm 0.01}$ | $0.48_{\pm 0.00}$ | $0.48_{\pm 0.00}$ | $0.47_{\pm 0.01}$ | $0.48_{\pm 0.02}$ | $0.49_{\pm 0.01}$ | $0.49_{\pm 0.01}$ |
| youtube | $0.88_{\pm 0.04}$ | $0.82_{\pm 0.02}$ | $0.89_{\pm 0.01}$ | $0.89_{\pm 0.01}$ | $0.87_{\pm 0.04}$ | $0.88_{\pm 0.03}$ | $0.89_{\pm 0.01}$ | $0.89_{\pm 0.01}$ |
| imdb | $0.50_{\pm 0.00}$ | $0.69_{\pm 0.00}$ | $0.74_{\pm 0.01}$ | $0.75_{\pm 0.00}$ | $0.59_{\pm 0.11}$ | $0.65_{\pm 0.12}$ | $0.70_{\pm 0.10}$ | $0.75_{\pm 0.00}$ |
| yelp | $0.80_{\pm 0.00}$ | $0.81_{\pm 0.00}$ | $0.80_{\pm 0.00}$ | $0.80_{\pm 0.00}$ | $0.77_{\pm 0.01}$ | $0.79_{\pm 0.02}$ | $0.81_{\pm 0.00}$ | $0.80_{\pm 0.01}$ |
| census | $0.54_{\pm 0.00}$ | $0.47_{\pm 0.00}$ | $0.52_{\pm 0.00}$ | $0.52_{\pm 0.00}$ | $0.50_{\pm 0.06}$ | $0.54_{\pm 0.01}$ | $0.54_{\pm 0.00}$ | $0.53_{\pm 0.01}$ |
| tennis | $0.77_{\pm 0.00}$ | $0.77_{\pm 0.00}$ | $0.77_{\pm 0.00}$ | $0.77_{\pm 0.00}$ | $0.59_{\pm 0.13}$ | $0.59_{\pm 0.13}$ | $0.64_{\pm 0.13}$ | $0.70_{\pm 0.11}$ |
| sms | $0.23_{\pm 0.01}$ | $0.66_{\pm 0.01}$ | $0.66_{\pm 0.01}$ | $0.23_{\pm 0.01}$ | $0.47_{\pm 0.21}$ | $0.55_{\pm 0.17}$ | $0.65_{\pm 0.05}$ | $0.73_{\pm 0.03}$ |
| cdr | $0.48_{\pm 0.00}$ | $0.35_{\pm 0.00}$ | $0.48_{\pm 0.00}$ | $0.50_{\pm 0.00}$ | $0.49_{\pm 0.01}$ | $0.49_{\pm 0.01}$ | $0.49_{\pm 0.01}$ | $0.49_{\pm 0.01}$ |
| basketball | $0.49_{\pm 0.02}$ | $0.49_{\pm 0.02}$ | $0.49_{\pm 0.02}$ | $0.44_{\pm 0.02}$ | $0.36_{\pm 0.15}$ | $0.41_{\pm 0.12}$ | $0.46_{\pm 0.02}$ | $0.46_{\pm 0.02}$ |
| spouse | $0.15_{\pm 0.00}$ | $0.35_{\pm 0.02}$ | $0.35_{\pm 0.02}$ | $0.37_{\pm 0.02}$ | $0.15_{\pm 0.00}$ | $0.15_{\pm 0.00}$ | $0.15_{\pm 0.00}$ | $0.19_{\pm 0.08}$ |
| commercial | $0.91_{\pm 0.00}$ | $0.78_{\pm 0.00}$ | $0.91_{\pm 0.00}$ | $0.91_{\pm 0.00}$ | $0.85_{\pm 0.05}$ | $0.84_{\pm 0.04}$ | $0.84_{\pm 0.04}$ | $0.85_{\pm 0.04}$ |

Now, we explore a different way of comparing models. Instead of making an explicit model selection, this experiment uses the proposed metrics, *i.e.*, accuracy lower/upper bounds, bounds average, to rank MLP classifiers. We rank the models using both the test set accuracy/F1 score (depending on Zhang et al. (2021)) and alternative metrics, *i.e.*, accuracy/F1 score lower/upper bounds, bounds average, label model, and small labeled sample sizes. Then, we calculate a Pearson correlation between rankings and display numbers in Table 4. If the numbers are higher, it means that the proposed selection method is capable of distinguishing good from bad models. Table 4 shows that the bounds average and label model methods usually return the best results when no labels are used. Moreover, in some cases using a small labeled sample for model selection can relatively hurt performance (when the validation set is small, there is a chance all models will have the same or similar performances, leading to smaller or null rank correlations).

Table 4: Ranking correlation when ranking using test set and alternative metric

| | Lower bound | Upper bound | Bounds avg | Label model | Labeled ($n = 10$) | Labeled ($n = 25$) | Labeled ($n = 50$) | Labeled ($n = 100$) |
|---|---|---|---|---|---|---|---|---|
| agnews | $0.90_{\pm 0.01}$ | $0.98_{\pm 0.01}$ | $0.93_{\pm 0.02}$ | $0.90_{\pm 0.01}$ | $0.89_{\pm 0.06}$ | $0.88_{\pm 0.05}$ | $0.87_{\pm 0.04}$ | $0.88_{\pm 0.07}$ |
| trec | $0.80_{\pm 0.03}$ | $0.96_{\pm 0.03}$ | $0.96_{\pm 0.05}$ | $0.88_{\pm 0.10}$ | $0.73_{\pm 0.37}$ | $0.87_{\pm 0.07}$ | $0.87_{\pm 0.09}$ | $0.82_{\pm 0.07}$ |
| semeval | $0.99_{\pm 0.01}$ | $0.81_{\pm 0.04}$ | $0.99_{\pm 0.01}$ | $0.99_{\pm 0.01}$ | $0.91_{\pm 0.06}$ | $0.95_{\pm 0.07}$ | $0.97_{\pm 0.02}$ | $0.95_{\pm 0.03}$ |
| chemprot | $0.76_{\pm 0.01}$ | $0.97_{\pm 0.02}$ | $0.77_{\pm 0.02}$ | $0.76_{\pm 0.01}$ | $0.90_{\pm 0.06}$ | $0.88_{\pm 0.07}$ | $0.91_{\pm 0.09}$ | $0.96_{\pm 0.05}$ |
| youtube | $0.95_{\pm 0.03}$ | $0.86_{\pm 0.08}$ | $0.95_{\pm 0.03}$ | $0.96_{\pm 0.02}$ | $0.83_{\pm 0.09}$ | $0.95_{\pm 0.04}$ | $0.93_{\pm 0.03}$ | $0.92_{\pm 0.03}$ |
| imdb | $-0.76_{\pm 0.00}$ | $0.76_{\pm 0.00}$ | $0.93_{\pm 0.05}$ | $0.94_{\pm 0.03}$ | $-0.09_{\pm 0.76}$ | $0.32_{\pm 0.75}$ | $0.68_{\pm 0.60}$ | $0.99_{\pm 0.01}$ |
| yelp | $0.92_{\pm 0.00}$ | $0.96_{\pm 0.02}$ | $0.97_{\pm 0.00}$ | $0.97_{\pm 0.00}$ | $0.82_{\pm 0.10}$ | $0.95_{\pm 0.05}$ | $0.99_{\pm 0.01}$ | $0.98_{\pm 0.02}$ |
| census | $0.98_{\pm 0.00}$ | $0.91_{\pm 0.00}$ | $0.95_{\pm 0.00}$ | $0.94_{\pm 0.00}$ | $0.57_{\pm 0.29}$ | $0.75_{\pm 0.02}$ | $0.75_{\pm 0.02}$ | $0.72_{\pm 0.02}$ |
| tennis | $0.89_{\pm 0.01}$ | $0.90_{\pm 0.01}$ | $0.89_{\pm 0.00}$ | $0.87_{\pm 0.01}$ | $0.09_{\pm 0.08}$ | $0.06_{\pm 0.15}$ | $0.12_{\pm 0.16}$ | $0.14_{\pm 0.23}$ |
| sms | $0.10_{\pm 0.06}$ | $0.99_{\pm 0.00}$ | $0.95_{\pm 0.01}$ | $0.89_{\pm 0.02}$ | $0.47_{\pm 0.38}$ | $0.58_{\pm 0.30}$ | $0.79_{\pm 0.02}$ | $0.80_{\pm 0.03}$ |
| cdr | $0.85_{\pm 0.00}$ | $0.95_{\pm 0.00}$ | $0.97_{\pm 0.00}$ | $1.00_{\pm 0.00}$ | $0.17_{\pm 0.21}$ | $0.26_{\pm 0.22}$ | $0.34_{\pm 0.18}$ | $0.36_{\pm 0.18}$ |
| basketball | $0.94_{\pm 0.00}$ | $0.30_{\pm 0.01}$ | $0.70_{\pm 0.00}$ | $0.85_{\pm 0.01}$ | $0.45_{\pm 0.37}$ | $0.60_{\pm 0.30}$ | $0.75_{\pm 0.00}$ | $0.75_{\pm 0.00}$ |
| spouse | $0.82_{\pm 0.00}$ | $0.88_{\pm 0.00}$ | $0.89_{\pm 0.00}$ | $0.96_{\pm 0.00}$ | $0.00_{\pm 0.00}$ | $0.00_{\pm 0.00}$ | $0.00_{\pm 0.00}$ | $0.04_{\pm 0.07}$ |
| commercial | $0.94_{\pm 0.00}$ | $0.98_{\pm 0.00}$ | $0.99_{\pm 0.00}$ | $1.00_{\pm 0.00}$ | $0.63_{\pm 0.17}$ | $0.71_{\pm 0.02}$ | $0.71_{\pm 0.02}$ | $0.70_{\pm 0.02}$ |

# B   A PROOF FOR THE DUALITY RESULT

*Proof of Theorem 2.1.* We start proving the result for $L$. See that

$$L = \inf_{\pi \in \Pi} \mathbb{E}_\pi[g(X, Y, Z)]$$

$$= \inf_{\{\pi_z \in \Pi_z\}_{z \in \mathcal{Z}}} \sum_{z \in \mathcal{Z}} \mathbb{P}(Z = z) \cdot \mathbb{E}_{\pi_z}[g(X, Y, Z) \mid Z = z]$$

$$= \sum_{z \in \mathcal{Z}} \mathbb{P}(Z = z) \cdot \inf_{\pi_z \in \Pi_z} \mathbb{E}_{\pi_z}[g(X, Y, Z) \mid Z = z]$$

with $\Pi_z$, is defined as

$$\Pi_z \triangleq \{\pi_z \in \Delta(\mathcal{X} \times \mathcal{Y}) : \pi_z \circ \rho_X^{-1} = P_{X|Z=z} \text{ and } \pi_z \circ \rho_Y^{-1} = P_{Y|Z=z}\}$$

That is, for each $z \in \mathcal{Z}$, $\Pi_z$ represents the set of couplings such that marginals are given by $P_{X|Z=z}$ and $P_{Y|Z=z}$. We can represent the problem in this way since the marginal distribution of $Z$ is fixed and, given that distribution, $\{\Pi_z\}$ specifies the same set of distributions as $\Pi$.

Realize that we have broken down our initial maximization problem in $|\mathcal{Z}|$ smaller minimization problems. Each of those minimization problems can be treated as an optimal transportation problem. Consequently, by Beiglböck & Schachermayer (2011, Theorem 1), for each $z \in \mathcal{Z}$, we get the following duality result

$$\inf_{\pi_z \in \Pi_z} \mathbb{E}_{\pi_z}[g(X, Y, Z) \mid Z = z] = \sup_{(\beta_z, \alpha_z) \in \Psi_z} \mathbb{E}[\beta_z(X) + \alpha_z(Y) \mid Z = z]$$

with

$$\Psi_z \triangleq \left\{ (\beta_z, \alpha_z) : \begin{array}{l} \beta_z : X \to [-\infty, \infty), \alpha_z : Y \to [-\infty, \infty) \\ \mathbb{E}[|\beta_z(X)| \mid Z = z] < \infty, \mathbb{E}[|\alpha_z(Y)| \mid Z = z] < \infty \\ \beta_z(x) + \alpha_z(y) \le g(x, y, z) \text{ for all } (x, y) \in X \times Y \end{array} \right\}$$

Moreover, partially optimizing on $\beta_z(x)$, we can set $\beta_z^*(x) = \min_{y \in \mathcal{Y}}[g(x, y, z) - \alpha_z(y)]$ and then

$$\inf_{\pi_z \in \Pi_z} \mathbb{E}_{\pi_z}[g(X, Y, Z) \mid Z = z] =$$
$$= \sup_{\alpha_z} \mathbb{E}\left[\min_{y \in \mathcal{Y}}[g(X, y, Z) + \alpha_z(y)] - \alpha_z(Y) \mid Z = z\right]$$

where $\alpha_z$ is a simple function taking values in the real line.

Consequently,

$$
\begin{aligned}
L &= \sum_{z \in \mathcal{Z}} \mathbb{P}(Z = z) \cdot \sup_{\alpha_z} \mathbb{E}\left[\min_{y \in \mathcal{Y}}[g(X, y, Z) + \alpha_z(y)] - \alpha_z(Y) \mid Z = z\right] \\
&= \sup_{\{\alpha_z\}_{z \in \mathcal{Z}}} \sum_{z \in \mathcal{Z}} \mathbb{P}(Z = z) \cdot \mathbb{E}\left[\min_{y \in \mathcal{Y}}[g(X, y, Z) + \alpha_z(y)] - \alpha_z(Y) \mid Z = z\right] \\
&= \sup_{\{\alpha_z\}_{z \in \mathcal{Z}}} \mathbb{E}\left[\min_{y \in \mathcal{Y}}[g(X, y, Z) + \alpha_Z(y)] - \alpha_Z(Y)\right]
\end{aligned}
$$

Because each $\alpha_z$ is a function assuming at most $|\mathcal{Y}|$ values and we have $|\mathcal{Z}|$ functions (one for each value of $z$), we can equivalently solve an optimization problem on $\mathbb{R}^{|\mathcal{Y}| \times |\mathcal{Z}|}$. Adjusting the notation,

$$
L = \sup_{a \in \mathbb{R}^{|\mathcal{Y}| \times |\mathcal{Z}|}} \mathbb{E}\left[\min_{\bar{y} \in \mathcal{Y}}\left[g(X, \bar{y}, Z) + a_{\bar{y}Z}\right]\right] - \mathbb{E}\left[a_{YZ}\right]
$$

From Beiglböck & Schachermayer (2011, Theorem 2), we know that the maximum is attained by some $a^* \in \mathbb{R}^{|\mathcal{Y}| \times |\mathcal{Z}|}$. To show that there is a maximizer in $\mathcal{A}$, we need to update the solution

$$
a^*_{\cdot z} \leftarrow a^*_{\cdot z} - \sum_{y \in \mathcal{Y}} a^*_{yz} \quad \text{(shift sub-vector by a constant)} \tag{B.1}
$$

for every $z \in \mathcal{Z}$. The objective function is not affected by such translations.

To prove the result for $U$, first realize that because $g$ is bounded, with no loss of generality, we can assume its range is a subset of $[0, 1]$. Define $c(x, y, z) = 1 - g(x, y, z)$ and see that

$$
U = \sup_{\pi \in \Pi} \mathbb{E}_\pi[1 - c(X, Y, Z)] = 1 - \inf_{\pi \in \Pi} \mathbb{E}_\pi[c(X, Y, Z)]
$$

Proceeding as before, we can obtain the final result by finding the dual formulation for $\inf_{\pi \in \Pi} \mathbb{E}_\pi[c(X, Y, Z)]$. $\qquad\square$

## C  PROOFS FOR THE ESTIMATION RESULTS

**We will analyze the estimator for $U$ (results for the estimator of $L$ can be obtained analogously).**

*Proof of Theorem 2.4.* By Lemmas C.1 and C.2, we can guarantee that $\mathbb{E}[\tilde{f}_{u,\varepsilon}(X, Z, a)]$ is minimized by a unique $a^*_{u,\varepsilon}$ (equalling it to $U_\varepsilon$) and that $\nabla^2_a \mathbb{E}[\tilde{f}_{u,\varepsilon}(X, Z, a^*_{u,\varepsilon})]$ is positive definite. Also, from the proof of Lemma C.2, we can see that $\tilde{f}_{u,\varepsilon}$ is convex in $a$ (because its Hessian is positive semidefinite). It is also true that the second moment of $\tilde{f}_{u,\varepsilon}(X, Z, a)$ is well defined (exists and finite) for each $a$ since $g$ is bounded. Define

$$\tilde{U}_\varepsilon \triangleq \inf_{a \in \mathbb{R}^{|\mathcal{Y}| \times |\mathcal{Z}|}} \frac{1}{n} \sum_{i=1}^n \tilde{f}_{u,\varepsilon}(X_i, Z_i, a)$$

and let $\tilde{a}_\varepsilon$ denote a value that attains that minimum, which is tight according to Lemma C.6. From Niemiro (1992, Theorem 4) and the conditions discussed above, we know that $\sqrt{n}(\tilde{a}_\varepsilon - a^*_{u,\varepsilon}) = \mathcal{O}_P(1)$. Then

$$
\begin{aligned}
&\sqrt{n}(\tilde{U}_\varepsilon - U_\varepsilon) \\
&= \frac{1}{\sqrt{n}} \left( \sum_i \tilde{f}_{u,\varepsilon}(X_i, Z_i, \tilde{a}_\varepsilon) - \sum_i \tilde{f}_{u,\varepsilon}(X_i, Z_i, a^*_{u,\varepsilon}) \right) + \sqrt{n} \left( \frac{1}{n} \sum_i \tilde{f}_{u,\varepsilon}(X_i, Z_i, a^*_{u,\varepsilon}) - U_\varepsilon \right) \\
&= \frac{1}{\sqrt{n}} \left( [\sqrt{n}(\tilde{a}_\varepsilon - a^*_{u,\varepsilon})]^\top [\frac{1}{n} \sum_i \nabla^2_a \tilde{f}_{u,\varepsilon}(X_i, Z_i, \bar{a})][\sqrt{n}(\tilde{a}_\varepsilon - a^*_{u,\varepsilon})] \right) + \\
&\quad + \sqrt{n} \left( \frac{1}{n} \sum_i \tilde{f}_{u,\varepsilon}(X_i, Z_i, a^*_{u,\varepsilon}) - U_\varepsilon \right) + o_P(1)
\end{aligned}
$$

(C.1)

where the first term is obtained by a second-order Taylor expansion of the summing functions around $\tilde{a}_\varepsilon$ ($\bar{a}$ is some random vector). Also, from the standard central limit theorem, we know that

$$\sqrt{n} \left( \frac{1}{n} \sum_i \tilde{f}_{u,\varepsilon}(X_i, Z_i, a^*_{u,\varepsilon}) - U_\varepsilon \right) \Rightarrow N(0, \operatorname{Var}\tilde{f}_{u,\varepsilon}(X, Z, a^*_{u,\varepsilon}))$$

Given that $\sqrt{n}(\tilde{a}_\varepsilon - a^*_{u,\varepsilon}) = \mathcal{O}_P(1)$ and that the Hessian has bounded entries (then $\mathcal{O}_P(1)$ as well), the first term in C.1 is $o_P(1)$. Because $a^*_{u,\varepsilon} \in \mathcal{A}$, we have that $f_{u,\varepsilon}(X, Z, a^*_{u,\varepsilon}) = \tilde{f}_{u,\varepsilon}(X, Z, a^*_{u,\varepsilon})$ and then

$$\sqrt{n}(\tilde{U}_\varepsilon - U_\varepsilon) \Rightarrow N(0, \operatorname{Var}f_{u,\varepsilon}(X, Z, a^*_{u,\varepsilon}))$$

by Slutsky's theorem. Since

$$\sqrt{n}(\hat{U}_\varepsilon - U) = \sqrt{n}(\hat{U}_\varepsilon - \tilde{U}_\varepsilon) + \sqrt{n}(\tilde{U}_\varepsilon - U_\varepsilon),$$

if we can show that

$$\sqrt{n}(\tilde{U}_\varepsilon - \hat{U}_\varepsilon) = o_P(1)$$

we are done.

Let $\hat{a}$ be solution for the problem in 2.5 and see that

$$|\tilde{U}_\varepsilon - \hat{U}_\varepsilon| =$$

$$= \left| \frac{1}{n} \sum_{i=1}^n \tilde{f}_{u,\varepsilon}(X_i, Z_i, \tilde{a}_\varepsilon) - \frac{1}{n} \sum_{i=1}^n \varepsilon \log \left[ \frac{1}{|\mathcal{Y}|} \sum_{y \in \mathcal{Y}} \exp \left( \frac{g(X_i, y, Z_i) + \hat{a}_{yZ_i}}{\varepsilon} \right) \right] + \mathbb{E}_{\hat{P}_{Y|Z}} [\hat{a}_{YZ} \mid Z = Z_i] \right|$$

$$\leq \left| \frac{1}{n} \sum_{i=1}^n \tilde{f}_{u,\varepsilon}(X_i, Z_i, \tilde{a}_\varepsilon) - \frac{1}{n} \sum_{i=1}^n \tilde{f}_{u,\varepsilon}(X_i, Z_i, \hat{a}) \right| +$$

$$+ \left| \frac{1}{n} \sum_{i=1}^n \tilde{f}_{u,\varepsilon}(X_i, Z_i, \hat{a}) - \frac{1}{n} \sum_{i=1}^n \varepsilon \log \left[ \frac{1}{|\mathcal{Y}|} \sum_{y \in \mathcal{Y}} \exp \left( \frac{g(X_i, y, Z_i) + \hat{a}_{yZ_i}}{\varepsilon} \right) \right] + \mathbb{E}_{\hat{P}_{Y|Z}} [\hat{a}_{YZ} \mid Z = Z_i] \right|$$

$$= \left( \frac{1}{n} \sum_{i=1}^n \tilde{f}_{u,\varepsilon}(X_i, Z_i, \hat{a}) - \frac{1}{n} \sum_{i=1}^n \tilde{f}_{u,\varepsilon}(X_i, Z_i, \tilde{a}_\varepsilon) \right) +$$

$$+ \left| \frac{1}{n} \sum_{i=1}^n \sum_y [\mathbb{P}(Y = y \mid Z = Z_i) - \hat{\mathbb{P}}(Y = y \mid Z = Z_i)] \hat{a}_{yZ_i} \right|$$

$$\leq \left( \frac{1}{n} \sum_{i=1}^n \tilde{f}_{u,\varepsilon}(X_i, Z_i, \hat{a}) - \frac{1}{n} \sum_{i=1}^n \varepsilon \log \left[ \frac{1}{|\mathcal{Y}|} \sum_{y \in \mathcal{Y}} \exp \left( \frac{g(X_i, y, Z_i) + \hat{a}_{yZ_i}}{\varepsilon} \right) \right] + \mathbb{E}_{\hat{P}_{Y|Z}} [\hat{a}_{YZ} \mid Z = Z_i] \right)$$

$$+ \left( \frac{1}{n} \sum_{i=1}^n \varepsilon \log \left[ \frac{1}{|\mathcal{Y}|} \sum_{y \in \mathcal{Y}} \exp \left( \frac{g(X_i, y, Z_i) + \tilde{a}_{\varepsilon yZ_i}}{\varepsilon} \right) \right] - \mathbb{E}_{\hat{P}_{Y|Z}} [\tilde{a}_{\varepsilon YZ} \mid Z = Z_i] - \frac{1}{n} \sum_{i=1}^n \tilde{f}_{u,\varepsilon}(X_i, Z_i, \tilde{a}_\varepsilon) \right)$$

$$+ \left| \frac{1}{n} \sum_{i=1}^n \sum_y [\mathbb{P}(Y = y \mid Z = Z_i) - \hat{\mathbb{P}}(Y = y \mid Z = Z_i)] \hat{a}_{yZ_i} \right|$$

$$= \left( \frac{1}{n} \sum_{i=1}^n \sum_y [\hat{\mathbb{P}}(Y = y \mid Z = Z_i) - \mathbb{P}(Y = y \mid Z = Z_i)] \hat{a}_{yZ_i} \right)$$

$$+ \left( \frac{1}{n} \sum_{i=1}^n \sum_y [\mathbb{P}(Y = y \mid Z = Z_i) - \hat{\mathbb{P}}(Y = y \mid Z = Z_i)] \tilde{a}_{\varepsilon yZ_i} \right)$$

$$+ \left| \frac{1}{n} \sum_{i=1}^n \sum_y [\mathbb{P}(Y = y \mid Z = Z_i) - \hat{\mathbb{P}}(Y = y \mid Z = Z_i)] \hat{a}_{yZ_i} \right|$$

$$\leq 2 \|\hat{a}\|_\infty \frac{1}{n} \sum_{i=1}^n \sum_y |\mathbb{P}(Y = y \mid Z = Z_i) - \hat{\mathbb{P}}(Y = y \mid Z = Z_i)| +$$

$$+ \|\tilde{a}_\varepsilon\|_\infty \frac{1}{n} \sum_{i=1}^n \sum_y |\mathbb{P}(Y = y \mid Z = Z_i) - \hat{\mathbb{P}}(Y = y \mid Z = Z_i)|$$

$$\leq 2 \|\hat{a}\|_\infty \sum_z \sum_y |\mathbb{P}(Y = y \mid Z = z) - \hat{\mathbb{P}}(Y = y \mid Z = z)| +$$

$$+ \|\tilde{a}_\varepsilon\|_\infty \sum_z \sum_y |\mathbb{P}(Y = y \mid Z = z) - \hat{\mathbb{P}}(Y = y \mid Z = z)|$$

$$\leq 4 \|\hat{a}\|_\infty \sum_z d_{\text{TV}} \left( \hat{P}_{Y|Z=z}, P_{Y|Z=z} \right) + 2 \|\tilde{a}_\varepsilon\|_\infty \sum_z d_{\text{TV}} \left( \hat{P}_{Y|Z=z}, P_{Y|Z=z} \right)$$

$$= \mathcal{O}_P(m^{-\lambda})$$

where the last equality is obtained using Assumption 2.3 and the conclusion of Lemma C.6. Consequently,

$$\sqrt{n}(\tilde{U}_\varepsilon - \hat{U}_\varepsilon) = \sqrt{n} \mathcal{O}_P(m^{-\lambda}) = o(m^\lambda) \mathcal{O}_P(m^{-\lambda}) = o_P(1)$$

Finally, using Slutsky's theorem,

$$\sqrt{n}(\hat{U}_\varepsilon - U) = \sqrt{n}(\hat{U}_\varepsilon - \tilde{U}_\varepsilon) + \sqrt{n}(\tilde{U}_\varepsilon - U) \Rightarrow N(0, \text{Var} f_{u,\varepsilon}(X, Z, a_{u,\varepsilon}^*))$$

$\square$

*Proof of Corollary 3.1.* We prove the asymptotic distribution of $\sqrt{n}(\hat{p}_{u,\varepsilon} - p_{u,\varepsilon})$. The result for the lower bound can be obtained analogously.

First, note that

$$\hat{\mathbb{P}}(h(X) = 1) - \mathbb{P}(h(X) = 1) = \mathcal{O}_P(m^{-1/2}) \tag{C.2}$$

by the standard central limit theorem.

Next, see that

$$
\begin{aligned}
\sqrt{n}\left(\hat{p}_{u,\varepsilon} - p_{u,\varepsilon}\right) = & \\
= & \sqrt{n}\left(\frac{\hat{U}_\varepsilon}{\hat{\mathbb{P}}(h(X)=1)} - \frac{U_\varepsilon}{\mathbb{P}(h(X)=1)}\right) \\
= & \frac{1}{\hat{\mathbb{P}}(h(X)=1)}\sqrt{n}\left(\hat{U}_\varepsilon - \frac{\hat{\mathbb{P}}(h(X)=1)}{\mathbb{P}(h(X)=1)}U_\varepsilon\right) \\
= & \frac{1}{\hat{\mathbb{P}}(h(X)=1)}\sqrt{n}\left(\hat{U}_\varepsilon - U_\varepsilon\right) + \frac{1}{\hat{\mathbb{P}}(h(X)=1)}\sqrt{n}\left(U_\varepsilon - \frac{\hat{\mathbb{P}}(h(X)=1)}{\mathbb{P}(h(X)=1)}U_\varepsilon\right) \\
= & \frac{1}{\hat{\mathbb{P}}(h(X)=1)}\sqrt{n}\left(\hat{U}_\varepsilon - U_\varepsilon\right) + U_\varepsilon\sqrt{n}\left(\frac{1}{\hat{\mathbb{P}}(h(X)=1)} - \frac{1}{\mathbb{P}(h(X)=1)}\right) \\
= & \frac{1}{\hat{\mathbb{P}}(h(X)=1)}\sqrt{n}\left(\hat{U}_\varepsilon - U_\varepsilon\right) + U_\varepsilon\sqrt{n}\left(\hat{\mathbb{P}}(h(X)=1) - \mathbb{P}(h(X)=1)\right)\left(\frac{-1}{\mathbb{P}(h(X)=1)^2}\right) + o_P\left(m^{-1/2}\right) \\
= & \frac{1}{\hat{\mathbb{P}}(h(X)=1)}\sqrt{n}\left(\hat{U}_\varepsilon - U_\varepsilon\right) + o_P(1) \\
\Rightarrow & N(0, \sigma_{p,u,\varepsilon}^2)
\end{aligned}
$$

where the (i) fifth and sixth lines equality is obtained using Taylor's theorem, (ii) sixth and seventh lines equality is obtained using observation C.2 and the fact that $n = o(m^{(2\lambda)\wedge 1})$, and (iii) seventh to eighth lines equality is obtained using observation C.2, Theorem 2.4, and Slutsky's theorem.

We prove the asymptotic distribution of $\sqrt{n}\left(\hat{r}_{u,\varepsilon} - r_{u,\varepsilon}\right)$. The result for the lower bound can be obtained analogously. From Lemma C.3, we know that there is an estimator $\hat{\mathbb{P}}(Y=1)$ such that $\hat{\mathbb{P}}(Y=1) - \mathbb{P}(Y=1) = \mathcal{O}_P(m^{-(\lambda\wedge 1/2)})$, *i.e.*, it has enough precision. We use that estimator.

Next, see that

$$
\begin{aligned}
\sqrt{n}\left(\hat{r}_{u,\varepsilon} - r_{u,\varepsilon}\right) = & \\
= & \sqrt{n}\left(\frac{\hat{U}_\varepsilon}{\hat{\mathbb{P}}(Y=1)} - \frac{U_\varepsilon}{\mathbb{P}(Y=1)}\right) \\
= & \frac{1}{\hat{\mathbb{P}}(Y=1)}\sqrt{n}\left(\hat{U}_\varepsilon - \frac{\hat{\mathbb{P}}(Y=1)}{\mathbb{P}(Y=1)}U_\varepsilon\right) \\
= & \frac{1}{\hat{\mathbb{P}}(Y=1)}\sqrt{n}\left(\hat{U}_\varepsilon - U_\varepsilon\right) + \frac{1}{\hat{\mathbb{P}}(Y=1)}\sqrt{n}\left(U_\varepsilon - \frac{\hat{\mathbb{P}}(Y=1)}{\mathbb{P}(Y=1)}U_\varepsilon\right) \\
= & \frac{1}{\hat{\mathbb{P}}(Y=1)}\sqrt{n}\left(\hat{U}_\varepsilon - U_\varepsilon\right) + U_\varepsilon\sqrt{n}\left(\frac{1}{\hat{\mathbb{P}}(Y=1)} - \frac{1}{\mathbb{P}(Y=1)}\right) \\
= & \frac{1}{\hat{\mathbb{P}}(Y=1)}\sqrt{n}\left(\hat{U}_\varepsilon - U_\varepsilon\right) + U_\varepsilon\sqrt{n}\left(\hat{\mathbb{P}}(Y=1) - \mathbb{P}(Y=1)\right)\left(\frac{-1}{\mathbb{P}(Y=1)^2}\right) + o_P\left(m^{-1/2}\right) \\
= & \frac{1}{\hat{\mathbb{P}}(Y=1)}\sqrt{n}\left(\hat{U}_\varepsilon - U_\varepsilon\right) + o_P(1) \\
\Rightarrow & N(0, \sigma_{r,u,\varepsilon}^2)
\end{aligned}
$$

Finally, we prove the asymptotic distribution of $\sqrt{n}\left(\hat{F}_{u,\varepsilon} - F_{u,\varepsilon}\right)$. The result for the lower bound can be obtained analogously. From the facts stated above, we know that

$$
\begin{aligned}
\hat{\mathbb{P}}(h(X)=1) + \hat{\mathbb{P}}(Y=1) - [\mathbb{P}(h(X)=1) + \mathbb{P}(Y=1)] = & \\
= [\hat{\mathbb{P}}(h(X)=1) - \mathbb{P}(h(X)=1)] + [\hat{\mathbb{P}}(Y=1) - \mathbb{P}(Y=1)] = & \\
= \mathcal{O}_P(m^{-(\lambda\wedge 1/2)}) &
\end{aligned}
$$

Then,

$$\sqrt{n}\left(\hat{F}_{u,\varepsilon} - F_{u,\varepsilon}\right) =$$

$$= \sqrt{n}\left(\frac{2\hat{U}_\varepsilon}{[\hat{\mathbb{P}}(h(X)=1)+\hat{\mathbb{P}}(Y=1)]} - \frac{2U_\varepsilon}{[\mathbb{P}(h(X)=1)+\mathbb{P}(Y=1)]}\right)$$

$$= \frac{2}{[\hat{\mathbb{P}}(h(X)=1)+\hat{\mathbb{P}}(Y=1)]}\sqrt{n}\left(\hat{U}_\varepsilon - \frac{[\hat{\mathbb{P}}(h(X)=1)+\hat{\mathbb{P}}(Y=1)]}{[\mathbb{P}(h(X)=1)+\mathbb{P}(Y=1)]}U_\varepsilon\right)$$

$$= \frac{2}{[\hat{\mathbb{P}}(h(X)=1)+\hat{\mathbb{P}}(Y=1)]}\sqrt{n}\left(\hat{U}_\varepsilon - U_\varepsilon\right) + \frac{1}{[\hat{\mathbb{P}}(h(X)=1)+\hat{\mathbb{P}}(Y=1)]}\sqrt{n}\left(2U_\varepsilon - \frac{[\hat{\mathbb{P}}(h(X)=1)+\hat{\mathbb{P}}(Y=1)]}{[\mathbb{P}(h(X)=1)+\mathbb{P}(Y=1)]}2U_\varepsilon\right)$$

$$= \frac{2}{[\hat{\mathbb{P}}(h(X)=1)+\hat{\mathbb{P}}(Y=1)]}\sqrt{n}\left(\hat{U}_\varepsilon - U_\varepsilon\right) + 2U_\varepsilon\sqrt{n}\left(\frac{1}{[\hat{\mathbb{P}}(h(X)=1)+\hat{\mathbb{P}}(Y=1)]} - \frac{1}{[\mathbb{P}(h(X)=1)+\mathbb{P}(Y=1)]}\right)$$

$$= \frac{2}{[\hat{\mathbb{P}}(h(X)=1)+\hat{\mathbb{P}}(Y=1)]}\sqrt{n}\left(\hat{U}_\varepsilon - U_\varepsilon\right) +$$

$$\quad + 2U_\varepsilon\sqrt{n}\left([\hat{\mathbb{P}}(h(X)=1)+\hat{\mathbb{P}}(Y=1)] - [\mathbb{P}(h(X)=1)+\mathbb{P}(Y=1)]\right)\left(\frac{-1}{[\mathbb{P}(h(X)=1)+\mathbb{P}(Y=1)]^2}\right) + o_P\left(m^{-1/2}\right)$$

$$= \frac{2}{[\hat{\mathbb{P}}(h(X)=1)+\hat{\mathbb{P}}(Y=1)]}\sqrt{n}\left(\hat{U}_\varepsilon - U_\varepsilon\right) + o_P(1)$$

$$\Rightarrow N(0, \sigma^2_{F,u,\varepsilon})$$

where all the steps are justified as before.

$\square$

### C.0.1 AUXILIARY LEMMAS

**Lemma C.1.** *Define*

$$\tilde{f}_{u,\varepsilon}(x,z,a) \triangleq f_{u,\varepsilon}(x,z,a) + \sum_{z'\in\mathcal{Z}}\left(\sum_{y\in\mathcal{Y}} a_{yz'}\right)^2$$

*Then*

$$\inf_{a\in\mathbb{R}^{|\mathcal{Y}|\times|\mathcal{Z}|}} \mathbb{E}[\tilde{f}_{u,\varepsilon}(X,Z,a)] = \inf_{a\in\mathcal{A}} \mathbb{E}[f_{u,\varepsilon}(X,Z,a)]$$

*Proof.* First, see that

$$\mathbb{E}[\tilde{f}_{u,\varepsilon}(X,Z,a)] \geq \mathbb{E}[f_{u,\varepsilon}(X,Z,a)]$$

From Lemma C.5, we know that there exists some $a^*_{u,\varepsilon} \in \mathbb{R}^{|\mathcal{Y}|\times|\mathcal{Z}|}$ such that

$$\inf_{a\in\mathcal{A}} \mathbb{E}[f_{u,\varepsilon}(X,Z,a)] = \mathbb{E}[f_{u,\varepsilon}(X,Z,a^*_{u,\varepsilon})]$$

For that specific $a^*_{u,\varepsilon}$, we have that

$$\mathbb{E}[\tilde{f}_{u,\varepsilon}(X,Z,a^*_{u,\varepsilon})] = \mathbb{E}[f_{u,\varepsilon}(X,Z,a^*_{u,\varepsilon})]$$

Consequently,

$$\inf_{a\in\mathbb{R}^{|\mathcal{Y}|\times|\mathcal{Z}|}} \mathbb{E}[\tilde{f}_{u,\varepsilon}(X,Z,a)] = \mathbb{E}[\tilde{f}_{u,\varepsilon}(X,Z,a^*_{u,\varepsilon})] = \mathbb{E}[f_{u,\varepsilon}(X,Z,a^*_{u,\varepsilon})] = \inf_{a\in\mathcal{A}} \mathbb{E}[f_{u,\varepsilon}(X,Z,a)]$$

$\square$

**Lemma C.2.** *The function of $a$ given by $\mathbb{E}[\tilde{f}_{u,\varepsilon}(X,Z,a)]$ has positive definite Hessian, i.e.,*

$$H_\varepsilon(a) = \nabla^2_a\mathbb{E}[\tilde{f}_{u,\varepsilon}(X,Z,a)] \succ 0$$

*and, consequently, it is strictly convex.*

*Proof.* See that

$$H_\varepsilon(a) = \nabla^2_a\mathbb{E}[f_{u,\varepsilon}(X,Z,a)] + \nabla^2_a\left[\sum_{z'\in\mathcal{Z}}\left(\sum_{y\in\mathcal{Y}} a_{yz'}\right)^2\right]$$

We start computing the first term in the sum.

First, for an arbitrary pair $(k, l) \in \mathcal{Y} \times \mathcal{Z}$, define

$$s_{kl}(x) \triangleq \frac{\exp\left(\frac{g(x,k,l)+a_{kl}}{\varepsilon}\right)}{\sum_y \exp\left(\frac{g(x,y,l)+a_{yl}}{\varepsilon}\right)}$$

Now, see that

$$\frac{\partial}{\partial a_{kl}} f_{u,\varepsilon}(x, z, a) = \mathbb{1}_{\{l\}}(z)\left[s_{kl}(x) - \mathbb{P}(Y = k \mid Z = l)\right]$$

and

$$\frac{\partial^2}{\partial a_{pl}\partial a_{kl}} f_{u,\varepsilon}(x, z, a) = \frac{1}{\varepsilon}\mathbb{1}_{\{l\}}(z)\left[\mathbb{1}_{\{k\}}(p)s_{kl}(x) - s_{kl}(x)s_{pl}(x)\right]$$

See that $\frac{\partial^2}{\partial a_{pb}\partial a_{kl}} f_{u,\varepsilon}(x, z, a) = 0$ if $b \neq l$. Consequently, the Hessian $\nabla_a^2 f_{u,\varepsilon}(x, z, a)$ is block diagonal.

Consequently, because the second derivatives are bounded, we can push them inside the expectations and get

$$\frac{\partial^2}{\partial a_{pl}\partial a_{kl}}\mathbb{E}\left[f_{u,\varepsilon}(X, Z, a)\right] =$$
$$= \mathbb{E}\left[\frac{\partial^2}{\partial a_{pl}\partial a_{kl}} f_{u,\varepsilon}(X, Z, a)\right]$$
$$= \frac{1}{\varepsilon}\mathbb{E}\left[\mathbb{1}_{\{l\}}(Z)\left[\mathbb{1}_{\{k\}}(p)s_{kl}(X) - s_{kl}(X)s_{pl}(X)\right]\right]$$
$$= \frac{1}{\varepsilon}\mathbb{1}_{\{k\}}(p) \cdot \mathbb{E}\left[\mathbb{P}(Z = l \mid X)s_{kl}(X)\right] - \frac{1}{\varepsilon}\mathbb{E}\left[\mathbb{P}(Z = l \mid X)s_{kl}(X)s_{pl}(X)\right]$$

Because $\nabla_a^2 f_{u,\varepsilon}(x, z, a)$ is block diagonal, we know that the Hessian

$$\nabla_a^2 \mathbb{E}\left[f_{u,\varepsilon}(X, Z, a)\right]$$

is block diagonal (one block for each segment $a_{\cdot z}$ of the vector $a$). Now, realize that

$$\nabla_a^2\left[\sum_{z' \in \mathcal{Z}}\left(\sum_{y \in \mathcal{Y}} a_{yz'}\right)^2\right]$$

is also block diagonal, with each block being matrices of ones. In this case, we also have one block for each segment $a_{\cdot z}$ of the vector $a$. Consequently, $H_\varepsilon(a)$ is block diagonal, and it is positive definite if and only if all of its blocks are positive definite. Let us analyse an arbitrary block of $H_\varepsilon(a)$, e.g., $\nabla_{a_{\cdot l}}^2 \mathbb{E}\left[\tilde{f}_{u,\varepsilon}(X, Z, a)\right]$. Let $\mathbf{s}_{\cdot l}(x)$ be the vector composed of $s_{kl}(x)$ for all $k$. If $\mathbb{1} \in \mathbb{R}^{|\mathcal{Y}|}$ denotes a vector of ones, then,

$$\nabla_{a_{\cdot l}}^2 \mathbb{E}\left[\tilde{f}_{u,\varepsilon}(X, Z, a)\right] =$$
$$= \frac{1}{\varepsilon}\mathrm{diag}\left(\mathbb{E}\left[\mathbb{P}(Z = l \mid X)\mathbf{s}_{\cdot l}(X)\right]\right) - \frac{1}{\varepsilon}\mathbb{E}\left[\mathbb{P}(Z = l \mid X)\mathbf{s}_{\cdot l}(X)\mathbf{s}_{\cdot l}(X)^\top\right] + \mathbb{1}\mathbb{1}^\top$$
$$= \mathbb{E}\left\{\frac{1}{\varepsilon}\mathbb{P}(Z = l \mid X)\left[\mathrm{diag}\left(\mathbf{s}_{\cdot l}(X)\right) - \mathbf{s}_{\cdot l}(X)\mathbf{s}_{\cdot l}(X)^\top + \frac{\varepsilon}{\mathbb{P}(Z=l)}\mathbb{1}\mathbb{1}^\top\right]\right\}$$
$$= \mathbb{E}\left\{\frac{1}{\varepsilon}\mathbb{P}(Z = l \mid X)\left[\mathrm{diag}\left(\mathbf{s}_{\cdot l}(X)\right) - \mathbf{s}_{\cdot l}(X)\mathbf{s}_{\cdot l}(X)^\top + \tilde{\mathbb{1}}\tilde{\mathbb{1}}^\top\right]\right\}$$

where $\tilde{\mathbb{1}} = \sqrt{\frac{\varepsilon}{\mathbb{P}(Z=l)}}\mathbb{1}$. See that $\mathrm{diag}\left(\mathbf{s}_{\cdot l}(x)\right)$ has rank $|\mathcal{Y}|$ (full rank) while $\mathbf{s}_{\cdot l}(x)\mathbf{s}_{\cdot l}(x)^\top$ is rank one for every $x \in \mathbb{R}^{d_X}$. Consequently, the rank of the difference

$$D(x) = \mathrm{diag}\left(\mathbf{s}_{\cdot l}(x)\right) - \mathbf{s}_{\cdot l}(x)\mathbf{s}_{\cdot l}(x)^\top$$

is greater or equal $|\mathcal{Y}| - 1$. It is the case that $\text{rank}(D(x)) = |\mathcal{Y}| - 1$ because $\tilde{\mathbb{1}}$ is in the null space of $D$:

$$D(x)\tilde{\mathbb{1}} = \left[\text{diag}\big(\mathbf{s}_{\cdot l}(x)\big) - \mathbf{s}_{\cdot l}(x)\mathbf{s}_{\cdot l}(x)^\top\right]\tilde{\mathbb{1}} = \sqrt{\frac{\varepsilon}{\mathbb{P}(Z=l)}}(\mathbf{s}_{\cdot l}(x) - \mathbf{s}_{\cdot l}(x)) = 0$$

Moreover, the range of $D(x)$ and $\tilde{\mathbb{1}}\tilde{\mathbb{1}}^\top$ are orthogonal. For any two vectors $\mathbf{v}, \mathbf{u} \in \mathbb{R}^{|\mathcal{Y}|}$, we have that

$$(D(x)\mathbf{v})^\top(\tilde{\mathbb{1}}\tilde{\mathbb{1}}^\top\mathbf{u}) = \mathbf{v}^\top D(x)^\top\tilde{\mathbb{1}}\tilde{\mathbb{1}}^\top\mathbf{u} = \mathbf{v}^\top D(x)\tilde{\mathbb{1}}\tilde{\mathbb{1}}^\top\mathbf{u} = \mathbf{v}^\top 0 \tilde{\mathbb{1}}^\top\mathbf{u} = 0$$

That implies $D(x) + \tilde{\mathbb{1}}\tilde{\mathbb{1}}^\top$ is full rank. To see that, let $\mathbf{v} \in \mathbb{R}^{|\mathcal{Y}|}$ be arbitrary and see

$$(D(x) + \tilde{\mathbb{1}}\tilde{\mathbb{1}}^\top)\mathbf{v} = 0 \Rightarrow D(x)\mathbf{v} = \tilde{\mathbb{1}}\tilde{\mathbb{1}}^\top\mathbf{v} = 0$$

Because $D(x)\mathbf{v} = 0$, it means that $\mathbf{v} = \theta\tilde{\mathbb{1}}$ for some constant $\theta$. If $\theta \neq 0$, then $\tilde{\mathbb{1}}\tilde{\mathbb{1}}^\top\mathbf{v} = \theta|\mathcal{Y}|\tilde{\mathbb{1}} \neq 0$. Therefore, $\theta = 0$ and $\mathbf{v} = 0$.

Now, let $\mathbf{u} \in \mathbb{R}^{|\mathcal{Y}|}$ be arbitrary non-null vector and see

$$\mathbf{u}^\top D(x)\mathbf{u} = \mathbf{u}^\top \text{diag}\big(\mathbf{s}_{\cdot l}(x)\big)\mathbf{u} - \mathbf{u}^\top \mathbf{s}_{\cdot l}(x)\mathbf{s}_{\cdot l}(x)^\top\mathbf{u}$$

$$= \sum_y u_y^2 s_{yl}(x) - \left(\sum_y u_y s_{yl}(x)\right)^2$$

$$= \left(\sum_y s_{yl}(x)\right)\left(\sum_y u_y^2 s_{yl}(x)\right) - \left(\sum_y u_y s_{yl}(x)\right)^2$$

$$= \left(\sum_y \sqrt{s_{yl}(x)}\sqrt{s_{yl}(x)}\right)\left(\sum_y \sqrt{u_y^2 s_{yl}(x)}\sqrt{u_y^2 s_{yl}(x)}\right) - \left(\sum_y u_y s_{yl}(x)\right)^2$$

$$\geq \left(\sum_y u_y s_{yl}(x)\right)^2 - \left(\sum_y u_y s_{yl}(x)\right)^2 = 0$$

by the Cauchy–Schwarz inequality. Then, $D(x)$ is positive semidefinite, and because $\tilde{\mathbb{1}}\tilde{\mathbb{1}}^\top$ is also positive semidefinite, their sum needs to be positive definite for all $x$ (that matrix is full rank). Each block of $H_\varepsilon(a)$ is positive definite; consequently, $H_\varepsilon(a)$ is positive definite. $\qquad\square$

**Lemma C.3.** *Assume Assumption 2.3 holds. Let*

$$\hat{\mathbb{P}}(Y = 1) = \tfrac{1}{m}\sum_{i=1}^m \mathbb{E}_{\hat{P}_{Y|Z}}[Y \mid Z = Z_i]$$

*Then*

$$\hat{\mathbb{P}}(Y = 1) - \mathbb{P}(Y = 1) = \mathcal{O}_P(m^{-(\lambda \wedge 1/2)})$$

*Proof.* To derive this result, see that

$$|\hat{\mathbb{P}}(Y = 1) - \mathbb{P}(Y = 1)| =$$

$$= \left|\tfrac{1}{m}\sum_{i=1}^m \mathbb{E}_{P_{Y|Z}}[Y \mid Z = Z_i] - \mathbb{P}(Y = 1) + \tfrac{1}{m}\sum_{i=1}^m \mathbb{E}_{\hat{P}_{Y|Z}}[Y \mid Z = Z_i] - \mathbb{E}_{P_{Y|Z}}[Y \mid Z = Z_i]\right|$$

$$\leq \left|\tfrac{1}{m}\sum_{i=1}^m \mathbb{E}_{P_{Y|Z}}[Y \mid Z = Z_i] - \mathbb{P}(Y = 1)\right| + \tfrac{1}{m}\sum_{i=1}^m \left|\mathbb{E}_{\hat{P}_{Y|Z}}[Y \mid Z = Z_i] - \mathbb{E}_{P_{Y|Z}}[Y \mid Z = Z_i]\right|$$

$$= \mathcal{O}_P(m^{-1/2}) + \tfrac{1}{m}\sum_{i=1}^m \left|\mathbb{P}_{\hat{P}_{Y|Z}}[Y = 1 \mid Z = Z_i] - \mathbb{P}_{P_{Y|Z}}[Y = 1 \mid Z = Z_i]\right|$$

$$\leq \mathcal{O}_P(m^{-1/2}) + \sum_{z\in\mathcal{Z}}\left|\mathbb{P}_{\hat{P}_{Y|Z}}[Y = 1 \mid Z = z] - \mathbb{P}_{P_{Y|Z}}[Y = 1 \mid Z = z]\right|$$

$$\leq \mathcal{O}_P(m^{-1/2}) + \sum_{y\in\{0,1\}}\sum_{z\in\mathcal{Z}}\left|\mathbb{P}_{\hat{P}_{Y|Z}}[Y = y \mid Z = z] - \mathbb{P}_{P_{Y|Z}}[Y = y \mid Z = z]\right|$$

$$= \mathcal{O}_P(m^{-1/2}) + 2\sum_{z\in\mathcal{Z}} d_{\text{TV}}\left(\hat{P}_{Y|Z=z}, P_{Y|Z=z}\right)$$

$$= \mathcal{O}_P(m^{-1/2}) + \mathcal{O}_P(m^{-\lambda})$$

$$= \mathcal{O}_P(m^{-(\lambda\wedge 1/2)})$$

where the standard central limit theorem obtains the third step, the sixth step is obtained by the formula of the total variation distance for discrete measures, and the seventh step is obtained by Assumption 2.3. $\qquad\square$

**Lemma C.4.** *For some $\kappa > 0$ and any $y \in \mathcal{Y}$ assume that $\kappa \leq p_y \leq 1 - \kappa$. Define $\mathcal{A} = \{a_y \in \mathbb{R} : \sum_y a_y = 0\}$. Then the optima of the following problems*

$$
\begin{aligned}
\inf_{a \in \mathcal{A}} f_u(a), \quad f_u(a) &\triangleq \mathbb{E}\Big[\epsilon \log \Big\{ \tfrac{1}{|\mathcal{Y}|} \textstyle\sum_y \exp\big(\tfrac{g(X,y)+a_y}{\epsilon}\big)\Big\}\Big] - \textstyle\sum_y p_y a_y\,, \\
\sup_{a \in \mathcal{A}} f_l(a), \quad f_l(a) &\triangleq \mathbb{E}\Big[-\epsilon \log \Big\{ \tfrac{1}{|\mathcal{Y}|} \textstyle\sum_y \exp\big(\tfrac{g(X,y)+a_y}{-\epsilon}\big)\Big\}\Big] - \textstyle\sum_y p_y a_y
\end{aligned}
\tag{C.3}
$$

*are attained in a compact set $K(\kappa, L) \subset \mathcal{A}$ where $\|g\|_\infty \leq L$.*

*Proof of lemma C.4.* We shall only prove this for the minimization problem. The conclusion for the maximization problem follows in a similar way.

**Strict convexity:** The second derivation of $f_u$ is

$$
\nabla_{a_y, a_{y'}} f_u(a) = \tfrac{1}{\epsilon}\mathbb{E}\big[p(X, y, a)\{\delta_{y,y'} - p(X, y', a)\}\big]
\tag{C.4}
$$

where $p(x, y, a) \triangleq \dfrac{\exp\big(\frac{g(X,y)+a_y}{\epsilon}\big)}{\sum_{i \in \mathcal{Y}} \exp\big(\frac{g(X,i)+a_i}{\epsilon}\big)}$. For any $u \in \mathbb{R}^{\mathcal{Y}}$ we have

$$
\begin{aligned}
u^\top \nabla^2 f_u(a) u &= \textstyle\sum_{i,j \in \mathcal{Y}} u_i u_j \nabla_{a_i, a_j} f_u(a) \\
&= \tfrac{1}{\epsilon} \textstyle\sum_{i,j \in \mathcal{Y}} u_i u_j \mathbb{E}\big[p(X, i, a)\{\delta_{i,j} - p(X, j, a)\}\big] \\
&= \tfrac{1}{\epsilon}\mathbb{E}\big[\textstyle\sum_i u_i^2 p(X, i, a) - \big\{\textstyle\sum_i u_i p(X, i, a)\big\}^2\big] = \tfrac{1}{\epsilon}\mathbb{E}[\sigma^2(X, u, a)] \geq 0
\end{aligned}
\tag{C.5}
$$

where $\sigma^2(x, u, a)$ is the variance of a categorical random variable taking the value $u_i$ with probability $p(x, i, a)$. This leads to the conclusion that the function is convex.

To establish strict convexity, we fix $a \in \mathcal{A}$ and notice that $0 < p(x, y, a) < 1$ (because $\|g\|_\infty \leq L$). Therefore, the $\sigma^2(x, u, a)$ can be zero only when the $u_i$'s are all equal. Since $\mathcal{A} = \{a \in \mathbb{R}^{\mathcal{Y}} : \sum_y a_y = 0\}$, such $u$ belongs to the space $\mathcal{A}$ in the unique case $u_i = 0$. This leads to the conclusion that for any $u \in \mathcal{A}$ and $u \neq 0$

$$
u^\top \nabla^2 f_u(a) u = \tfrac{1}{\epsilon}\mathbb{E}[\sigma^2(X, u, a)] > 0\,.
$$

Therefore, $f_u$ is strictly convex and has a unique minimizer in $\mathcal{A}$. In the remaining part of the proof, we focus on the first-order condition.

**First order condition:** The first-order condition is

$$
h(a, y) = p_y \text{ for all } y \in \mathcal{Y}, \text{ where } h(a, y) \triangleq \mathbb{E}[p(X, y, a)]
\tag{C.6}
$$

To show that equation C.6 has a solution in a compact ball $K(\epsilon, \kappa, L) \triangleq \{a \in \mathcal{A} : \|a\|_2 \leq M(\epsilon, \kappa, L)\}$ we construct an $M(\epsilon, \kappa, L) > 0$ such that $\min_y h(a, y) < \kappa$ whenever $\|a\|_2 > M(\epsilon, \kappa, L)$. Since $\kappa \leq p_y \leq 1 - \kappa$ for all $y \in \mathcal{Y}$ this concludes that the solution to equation C.6 must be inside the $K(\epsilon, \kappa, L)$.

**Construction of the compact set:** Let us define $y_{\min} \triangleq \arg\min_y a_y$ and $y_{\max} \triangleq \arg\max_y a_y$. To construct $M(\kappa, L, \epsilon)$ (we shall write it simply as $M$ whenever it is convenient to do so) we notice that

$$
\begin{aligned}
h(a, y_{\min}) &= \mathbb{E}[p(X, y_{\min}, a)] \\
&= \mathbb{E}\Bigg[\frac{\exp\big(\frac{g(X,1)+a_{y_{\min}}}{\epsilon}\big)}{\sum_{i \in \mathcal{Y}} \exp\big(\frac{g(X,i)+a_i}{\epsilon}\big)}\Bigg] \\
&\leq \frac{\exp\big(\frac{L+a_{y_{\min}}}{\epsilon}\big)}{\exp\big(\frac{L+a_{y_{\min}}}{\epsilon}\big) + \sum_{i \geq 2} \exp\big(\frac{-L+a_i}{\epsilon}\big)} \\
&= \frac{\exp\big(\frac{2L}{\epsilon}\big)}{\exp\big(\frac{2L}{\epsilon}\big) + \sum_{i \geq 2} \exp\big(\frac{a_i - a_{y_{\min}}}{\epsilon}\big)} \\
&\leq \frac{\exp\big(\frac{2L}{\epsilon}\big)}{\exp\big(\frac{2L}{\epsilon}\big) + \exp\big(\frac{a_{y_{\max}} - a_{y_{\min}}}{\epsilon}\big)} \leq \frac{\exp\big(\frac{2L}{\epsilon}\big)}{\exp\big(\frac{2L}{\epsilon}\big) + \exp\big(\frac{R}{\epsilon}\big)}\,,
\end{aligned}
\tag{C.7}
$$

where
$$R \triangleq \min\{\max_y a_y - \min_y a_y : \textstyle\sum_i a_i = 0, \sum_i a_i^2 > M^2\}.$$

We rewrite the constraints of the optimization as:
$$R \triangleq \min\left\{\max_y a_y - \min_y a_y : \mathrm{mean}\{a_i\} = 0, \mathrm{var}\{a_i\} > \tfrac{M^2}{|\mathcal{Y}|}\right\}.$$

where we use the Popoviciu's inequality on variance to obtain
$$\tfrac{M^2}{|\mathcal{Y}|} < \mathrm{var}\{a_i\} \le \tfrac{(\max_y a_y - \min_y a_y)^2}{4}, \quad \text{or} \quad \max_y a_y - \min_y a_y > \tfrac{2M}{\sqrt{|\mathcal{Y}|}},$$

Thus $R \ge \frac{2M}{\sqrt{|\mathcal{Y}|}}$ whenever $\|a\|_2 > M$. We use this inequality in equation C.7 and obtain
$$h(a, y_{\min}) \le \frac{\exp\left(\frac{2L}{\epsilon}\right)}{\exp\left(\frac{2L}{\epsilon}\right) + \exp\left(\frac{R}{\epsilon}\right)} \le \frac{\exp\left(\frac{2L}{\epsilon}\right)}{\exp\left(\frac{2L}{\epsilon}\right) + \exp\left(\frac{2M}{\epsilon\sqrt{|\mathcal{Y}|}}\right)}.$$

Finally, we choose $M = M(\epsilon, \kappa, L) > 0$ large enough such that
$$h(a, y_{\min}) \le \frac{\exp\left(\frac{2L}{\epsilon}\right)}{\exp\left(\frac{2L}{\epsilon}\right) + \exp\left(\frac{2M}{\epsilon\sqrt{|\mathcal{Y}|}}\right)} < \kappa.$$

For such an $M$ we concludes that $h(a, y_{\min}) < \kappa$ whenever $\|a\|_2 > M$.

$\square$

**Lemma C.5.** *$L_\varepsilon$ and $U_\varepsilon$ are attained by some optimizers in a compact set $K(\epsilon, \kappa_z, z \in \mathcal{Z}; L) \supset \mathcal{A}$ (equation 2.3), where $\kappa_z = \min\{p_{y|z}, 1 - p_{y|z} : y \in \mathcal{Y}\}$.*

*Proof of lemma C.5.* We shall only prove the case of the minimization problem. The proof for the maximization problem uses a similar argument.

A decomposition of the minimization problem according to the values of $Z$ follows.

$$\min_{a \in \mathcal{A}} \mathbb{E}\left[\epsilon \log\left\{\tfrac{1}{|\mathcal{Y}|}\sum_{y \in \mathcal{Y}} \exp\left(\tfrac{g(X,y,Z) + a_{Y,Z}}{\epsilon}\right)\right\} - \mathbb{E}\left[a_{Y,Z} \mid Z\right]\right]$$

$$= \sum_z p_z \left\{ \min_{\substack{a_{\cdot,z} \in \mathbb{R}^{\mathcal{Y}} \\ \sum_y a_{y,z} = 0}} \mathbb{E}\left[\epsilon \log\left\{\tfrac{1}{|\mathcal{Y}|}\sum_{y \in \mathcal{Y}} \exp\left(\tfrac{g(X,y,z) + a_{Y,z}}{\epsilon}\right)\right\} \mid Z = z\right] - \mathbb{E}[a_{Y,z} \mid Z = z] \right\}$$

$$= \sum_z p_z \left\{ \min_{\substack{a_{\cdot,z} \in \mathbb{R}^{\mathcal{Y}} \\ \sum_y a_{y,z} = 0}} \mathbb{E}\left[\epsilon \log\left\{\tfrac{1}{|\mathcal{Y}|}\sum_{y \in \mathcal{Y}} \exp\left(\tfrac{g(X,y,z) + a_{Y,z}}{\epsilon}\right)\right\} \mid Z = z\right] - \sum_y a_{y,z} p_{y|z} \right\}$$

$$\tag{C.8}$$

where $p_z = P(Z = z)$ and $p_{y|z} = P(Y = y \mid Z = z)$. We fix a $z$ and consider the corresponding optimization problem in the decomposition. Then according to lemma C.4 the optimal point is in a compact set $K(\epsilon, \kappa_z, L)$. Thus the optimal point of the full problem is in the Cartesian product $\prod_z K(\epsilon, \kappa_z, L) \subset \mathcal{A}$, which a compact set. Thus we let $K(\epsilon, \kappa_z, z \in \mathcal{Z}; L) = \prod_z K(\epsilon, \kappa_z, L)$.

$\square$

**Lemma C.6.** *The optimizers for the problems in equation 2.5 are tight with respect to $m, n \to \infty$.*

*Proof of the lemma C.6.* Let $p_{y|z} = P(Y = y \mid Z = z)$ and $\hat{p}_{y|z}^{(m)} = \hat{P}_{Y|Z=z}^{(m)}(y)$. Since
$$\tfrac{1}{2}\sum_y |\hat{p}_{y|z}^{(m)} - p_{y|z}| = d_{\mathrm{TV}}\left(\hat{P}_{Y|Z=z}^{(m)}, P_{Y|Z=z}\right) = \mathcal{O}_P(m^{-\lambda})$$

according to the assumption 2.3, for sufficiently large $m$ with high probability $p(m)$ ($p(m) \to 1$ as $m \to \infty$) $\tfrac{\kappa_z}{2} \le \hat{p}_{y|z}^{(m)} \le 1 - \tfrac{\kappa_z}{2}$ for all $y$. Fix such an $m$ and $\hat{p}_{y|z}^{(m)}$. In lemma C.5 we replace $P_{X,Z}$

with $\sum_{i=1}^{n} \delta_{X_i, Z_i}$ and $p_{y|z}$ with $\hat{p}_{y|z}^{(m)}$ to reach the conclusion that the optimizer is in the compact set $K(\epsilon, {}^{\kappa_z}\!/2, z \in \mathcal{Z}; L)$. Thus, for sufficiently large $m$ and any $n$

$$\mathbb{P}\big(\text{optimizer is in } K(\epsilon, {}^{\kappa_z}\!/2, z \in \mathcal{Z}; L)\big) \geq p(m) \,.$$

This establishes that the optimizers are tight.

$\square$

# D MORE ON EXPERIMENTS

## D.1 EXTRA RESULTS FOR THE WRENCH EXPERIMENT

Extra results for binary classification datasets can be found in Figures 4, 5, 6. In Figure 5, we truncate the number of weak labels to be at most 10 in all datasets (in order to use the "Oracle" approach). In Figure 6, we do not use the "Oracle" approach, then using all weak labels. However, We can see that when the number of weak labels is large, estimating the label model is harder, potentially leading to invalid bounds, *e.g.*, see "census" dataset.

In Tables 5 and 6, we can see the results for multinomial classification datasets. In Figure 5, we truncate the number of weak labels to be at most 10 in all datasets (in order to use the "Oracle" approach) while in the second table, we do not. In general, we can see that Snorkel is a better label model when compared with FlyingSquid, in the sense that the bounds contain the test score more frequently.

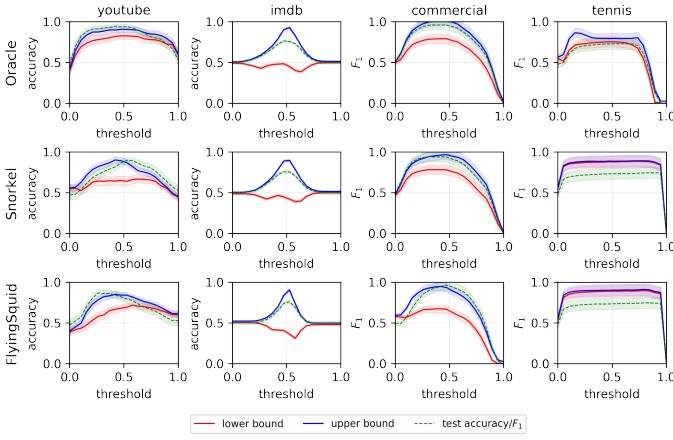

Figure 4: Bounds on classifier accuracies across classification thresholds for the Wrench datasets. Despite potential misspecification in Snorkel's and FlyingSquid's label model, it performs comparably to using labels to estimate $P_{Y|Z}$, giving approximate but meaningful bounds. .

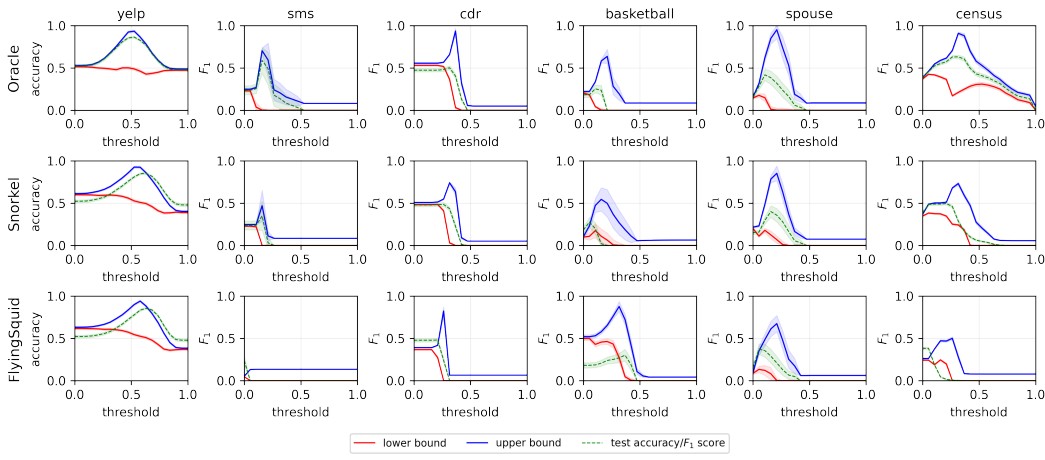

Figure 5: Bounds on classifier accuracies and F1 scores across classification thresholds for the Wrench datasets (truncating the number of weak labels in 10). Despite potential misspecification in Snorkel's and FlyingSquid's label model, it performs comparably to using labels to estimate $P_{Y|Z}$, giving approximate but meaningful bounds.

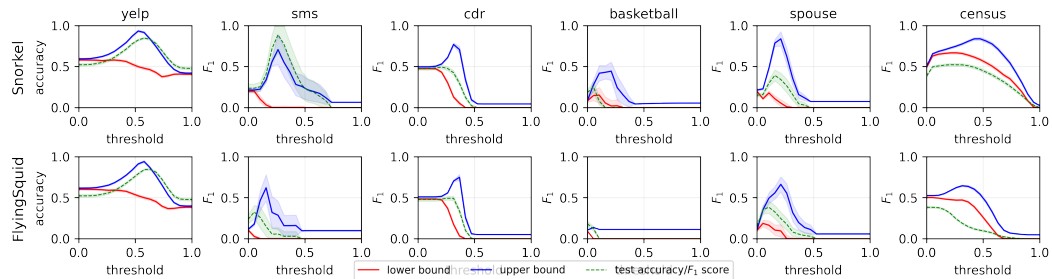

Figure 6: Bounds on classifier accuracies and F1 scores across classification thresholds for the Wrench datasets (using the full set of weak labels). Despite potential misspecification in Snorkel's and FlyingSquid's label model, it performs comparably to using labels to estimate $P_{Y|Z}$, giving approximate but meaningful bounds.

Table 5: Bounding the accuracy of classifiers in multinomial classification (truncating the number of weak labels in 10)

| Dataset | Label model | Lower bound | Upper bound | Test accuracy |
|---------|-------------|-------------|-------------|---------------|
| agnews | Oracle | $0.49_{\pm 0.01}$ | $0.95_{\pm 0.01}$ | $0.82_{\pm 0.01}$ |
| | Snorkel | $0.45_{\pm 0.01}$ | $0.9_{\pm 0.01}$ | $0.8_{\pm 0.01}$ |
| | FlyingSquid | $0.14_{\pm 0.0}$ | $0.61_{\pm 0.01}$ | $0.79_{\pm 0.01}$ |
| trec | Oracle | $0.04_{\pm 0.02}$ | $0.85_{\pm 0.03}$ | $0.38_{\pm 0.04}$ |
| | Snorkel | $0.01_{\pm 0.01}$ | $0.67_{\pm 0.03}$ | $0.3_{\pm 0.04}$ |
| | FlyingSquid | $0.16_{\pm 0.0}$ | $0.18_{\pm 0.0}$ | $0.28_{\pm 0.04}$ |
| semeval | Oracle | $0.24_{\pm 0.02}$ | $0.38_{\pm 0.02}$ | $0.36_{\pm 0.04}$ |
| | Snorkel | $0.3_{\pm 0.0}$ | $0.31_{\pm 0.0}$ | $0.32_{\pm 0.04}$ |
| | FlyingSquid | $0.12_{\pm 0.0}$ | $0.14_{\pm 0.0}$ | $0.32_{\pm 0.04}$ |
| chemprot | Oracle | $0.22_{\pm 0.02}$ | $0.59_{\pm 0.02}$ | $0.41_{\pm 0.02}$ |
| | Snorkel | $0.14_{\pm 0.02}$ | $0.67_{\pm 0.01}$ | $0.39_{\pm 0.02}$ |
| | FlyingSquid | $0.01_{\pm 0.0}$ | $0.29_{\pm 0.01}$ | $0.41_{\pm 0.02}$ |

## D.2 EXTRA PLOTS FOR THE HATE SPEECH DETECTION EXPERIMENT

In Table 7, we can see the same results already in the main text plus the results for FlyingSquid.

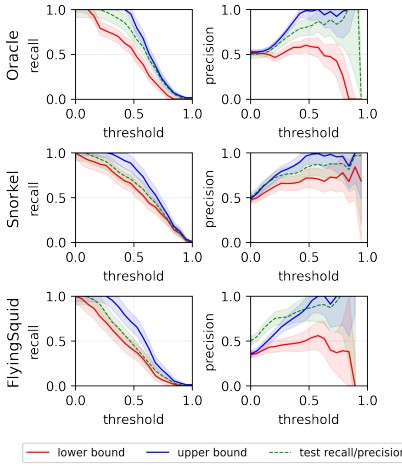

Figure 7: Precision and recall bounds for hate speech detection. These plots guide practitioners to trade off recall and precision in the absence of high-quality labels.

Table 6: Bounding the accuracy of classifiers in multinomial classification (using all weak labels)

| Dataset | Label model | Lower bound | Upper bound | Test accuracy |
|---------|-------------|-------------|-------------|---------------|
| agnews | Snorkel | $0.45_{\pm 0.01}$ | $0.9_{\pm 0.01}$ | $0.79_{\pm 0.01}$ |
| | FlyingSquid | $0.14_{\pm 0.0}$ | $0.61_{\pm 0.01}$ | $0.79_{\pm 0.01}$ |
| trec | Snorkel | $0.33_{\pm 0.04}$ | $0.67_{\pm 0.03}$ | $0.49_{\pm 0.04}$ |
| | FlyingSquid | $0.04_{\pm 0.02}$ | $0.31_{\pm 0.02}$ | $0.24_{\pm 0.04}$ |
| semeval | Snorkel | $0.34_{\pm 0.03}$ | $0.69_{\pm 0.03}$ | $0.53_{\pm 0.04}$ |
| | FlyingSquid | $0.12_{\pm 0.0}$ | $0.14_{\pm 0.0}$ | $0.32_{\pm 0.04}$ |
| chemprot | Snorkel | $0.36_{\pm 0.02}$ | $0.73_{\pm 0.02}$ | $0.48_{\pm 0.02}$ |
| | FlyingSquid | $0.04_{\pm 0.0}$ | $0.22_{\pm 0.01}$ | $0.45_{\pm 0.02}$ |

## D.3 EXAMPLES OF PROMPTS USED IN SECTION 4.2

### Prompt 1

```
You should classify the target sentence as "spam" or "ham". If          1
   definitions or examples are introduced, you should consider them when
    classifying sentences. Respond with "spam" or "ham".
                                                                         2
Target sentence: if your like drones, plz subscribe to Kamal Tayara. He  3
   takes videos with  his drone that are absolutely beautiful. --
   Response:
```

### Prompt 2

```
You should classify the target sentence as "spam" or "ham". If          1
   definitions or examples are introduced, you should consider them when
    classifying sentences. Respond with "spam" or "ham".
                                                                         2
Definition of spam: spam is a term referencing a broad category of       3
   postings which abuse web-based forms to post unsolicited
   advertisements as comments on forums, blogs, wikis and online
   guestbook.
                                                                         4
Definition of ham: texts that are not spam.                              5
                                                                         6
Target sentence: if your like drones, plz subscribe to Kamal Tayara. He  7
   takes videos with  his drone that are absolutely beautiful. --
   Response:
```

### Prompt 3

```
You should classify the target sentence as "spam" or "ham". If          1
   definitions or examples are introduced, you should consider them when
    classifying sentences. Respond with "spam" or "ham".
                                                                         2
Example 0: 860,000,000 lets make it first female to reach one billion!!  3
   Share it and replay it!  -- Response: ham
                                                                         4
Example 1: Waka waka eh eh -- Response: ham                              5
                                                                         6
Example 2: You guys should check out this EXTRAORDINARY website called   7
   ZONEPA.COM . You can make money online and start working from home
   today as I am! I am making over \$3,000+ per month at ZONEPA.COM !
   Visit Zonepa.com and check it out! How does the mother approve the
   axiomatic insurance? The fear appoints the roll. When does the space
   prepare the historical shame? -- Response: spam
                                                                         8
Example 3: Check out  these Irish guys cover  of Avicii\&\#39;s  Wake Me  9
   Up!  Just search...  \"wake me up Fiddle Me Silly\" Worth a
    listen  for the gorgeous fiddle player! -- Response: spam
```

```
Example 4: if you want to win money at hopme click here <a href="https://
    www.paidverts.com/ref/sihaam01">https://www.paidverts.com/ref/
    sihaam01</a> it\&\#39;s work 100/100 -- Response: spam

Target sentence: if your like drones, plz subscribe to Kamal Tayara. He
    takes videos with  his drone that are absolutely beautiful. --
    Response:
```