# OpenReview forum: "Estimating Fréchet bounds for validating programmatic weak supervision"
_ICLR.cc/2024/Conference — Submitted to ICLR 2024_

### Official Review · Reviewer_JyZF · 2023-10-27

**Soundness:** 3 good
**Presentation:** 4 excellent
**Contribution:** 3 good
**Rating:** 5
**Confidence:** 4

**Summary:**

This paper proposes solutions via convex programs to estimate Frechet bounds for Programmatic Weak Supervision (PWS). This approach uses estimates of the true labels via labelmodels (i.e., different aggregation schemes that exist in the literature). With these estimates of the labels, they provide an approach to estimate bounds on the accuracy (and other quantities) of the weak labelers. They provide experiments to check the validity of their bounds and also provide experiments with weak labelers generated via prompting to examine how their bounds perform under instances of weak labelers with different qualities/accuracies.

**Strengths:**

1. The authors provide a nice analysis of their estimation scheme.

2. They derive the asymptotic distributions of their estimators and show they are consistent.

3. They also provide CI for their estimates given finite sample data.

4. A novel application of Frechet bounds (and the corresponding literature) to the field of programmatic weak supervision.

**Weaknesses:**

1. One weakness is that this approach is fundamentally reliant on the quality of the label model. This is manifested in assumption 2.3, which states that the estimate of the conditional distribution of $Y | Z$ should approach the true conditional distribution. However, given biased and underperforming weak labelers, does this assumption actually hold? As such, are these truly valid bounds? They are violated in a few instances in Figure 1 (lower bound in second to the left in the bottom row, upper bound in second to left in the top row).

2. As highlighted by the authors in their Limitations section in the conclusion, this approach implicitly makes the same assumptions as the specified labelmodel, which can indeed be making assumptions about the conditional independence of the weak labelers (e.g., what is default in the Snorkel implementation).

3. I’m a bit confused as to what is the underlying message of Section 4.3. It seems to me that the proposed bounds perform poorly in the case with a few LLM generated weak labelers. I currently am understanding the “label set selection” approach to be picking weak labelers based on “accuracy”, except where accuracy is measured by disagreement with the estimated weak labels (i.e., label model outputs). As such, this would simply be pruning away weak labelers that disagree with the labelmodel, which seems somewhat undesirable.

4. There seems to be a lack of discussion about existing work on adversarial methods in PWS (namely [1] and subsequent followup work). I realize this existing work focuses on the classifier error rate, although the relaxed convex program approach seems similar.

[1] Arachie, Chidubem, and Bert Huang. "Adversarial label learning." Proceedings of the AAAI Conference on Artificial Intelligence. Vol. 33. No. 01. 2019.

**Questions:**

Primarily my first point in the weaknesses section above: is this assumption too strong? Given imperfect weak labelers, we cannot correctly estimate the true label (with a labelmodel), and, thus, wouldn’t these bounds would be violated in many cases? Any justification about the strength of these assumptions (i.e., allowing to swap in label model estimates with true labels) would be much appreciated.

---

> ### Author Response · Authors · 2023-11-20
>
> **On the quality of weak labels:** The quality of weak labels is usually **not** a problem for the label model estimation. According to [2], "a sufficient condition (for the identification of the label model) is the standard one of assuming non-adversarial sources, i.e., that all sources have greater (accuracy) than random accuracy". Consequently, our bounds should be valid asymptotically if all weak labelers have non-adversarial accuracy (weak assumption) and the label model is correctly specified. However, the quality of the weak labels will influence how good the weakly supervised classifier is and the size of the gap between the bounds $L$ and $U$ (see "When are the Fréchet bounds informative?" in the general comments above).
>
> **Label model misspecification:** If the label model is misspecified, e.g., by assuming weak labelers are conditionally independent given $Y$ (like in Snorkel), the resulting bounds may not be valid. We have studied this issue in the new version of the paper and included in Section A.3 a new result where assumption 2.3 does not hold. Please find a more detailed response to this issue in the general comments above.
>
> **Comment about label set selection:** In Figure 3, accuracy is **not** related to the alignment of weak labels and the label model. In fact, accuracy here is related to the performance of the weakly supervised classifier. For each one of the three plots, we have different sets of weak labels, and for each one of these sets, we estimate a different label model and use that label model to estimate the performance bounds. It is clear from the second and third plots that the third set of weak labels is better than the second because the red curve in the third plot is above the blue line in the second. In other words, the worst-case scenario using the third set of weak labels is better than the best-case scenario using the second set of weak labels. In this example, we are basically doing model selection. Please check Section A.4 for more details about model selection using our performance bounds.
>
> **Related work:** Thank you for your suggestions for new references. Adversarial learning is indeed related to our work in some ways (but not a substitute). We will update the "Related work" section and incorporate your suggested references in the final version of this paper.
>
> ## References
>
> [2] Ratner, A., Hancock, B., Dunnmon, J., Sala, F., Pandey, S., & Ré, C. (2019, July). Training complex models with multi-task weak supervision. In Proceedings of the AAAI Conference on Artificial Intelligence (Vol. 33, No. 01, pp. 4763-4771).

---

> > ### Comment · Reviewer_JyZF · 2023-11-22
> > **Reviewer Response**
> >
> > Thanks to the authors for their detailed response and additional proofs and experiments.
> >
> > I appreciate the clarifications regarding label model misspecification and label set selection. However, I still have remaining concerns:
> >
> > 1. I understand now that the violation of the bounds in the figures is due to the misspecification of the labelmodel (or perhaps the rare case for worse than random weak labelers). However, I feel as though the results regarding misspecification are not useful from a pratical standpoint, as the notion of the TV distance is incomputable in practice. Therefore, there’s no way to implement such results into a framework to get truly valid bounds (given misspecification).
> >
> > 2. Label set selection seems to be weirdly motivated in this setting for me. Given a correctly specified label model and a better than random weak labeler (which is exactly the setting that where this method achieves valid bounds), then there is no need to prune away weak labelers as they are helpful for the labelmodel. I agree that removing these low-accuracy weak labelers can be helpful given misspecification or a violation of assumptions (which is studied in [1]), but in the setting where this paper achieves valid bounds, then I believe there is no need for this pruning.
> >
> > As such, I opt to retain my score.
> >
> > [1]  Mazzetto, A., et. al. Semi-supervised aggregation of dependent weak supervision sources with performance guarantees.

---

### Official Review · Reviewer_JZxR · 2023-11-01

**Soundness:** 4 excellent
**Presentation:** 4 excellent
**Contribution:** 3 good
**Rating:** 6
**Confidence:** 3

**Summary:**

The paper develops an algorithm for estimating the Frechet bounds based on convex programs. The paper also provides an asymptotic theoretical guarantee of the proposed estimator. Finally, the paper provides experiments showing a case that the bound can be used to evaluate classifiers in the weak supervision setting without access to labels.

**Strengths:**

1. The proposed estimator also has a theoretical guarantee.
2. The experiment demonstrates that the bound is meaningful.
3. The mathematical parts are rigorous.

**Weaknesses:**

1. The estimator in equation (2.4) heavily relies on the distribution P_{Y\mid Z} which is not observed. The paper suggested using a label model to approximate such distribution. This requires the label model to be “good”. However, in this case, we can just use it to evaluate classifiers directly without deriving the Frechet bound.

2. For most tasks, a small set of labeled data is available. In fact, snorkel also relies on a small dev set to tune the hyperparameters of the final predictor. One could also use these labels to provide a bound on the performance of a classifier. It would be interesting to compare a confidence interval from a small dev set and the Frechet bound.

**Questions:**

see above.

---

> ### Author Response · Authors · 2023-11-20
>
> **On using the label model solely to evaluate classifiers:**  As we discuss in the second paragraph of our "Introduction" section (specifically, in the last eight lines of page 1), the classifier cannot be evaluated even when the label model $P_{Y \mid Z}$ is perfectly known. This is because the joint distribution $P_{X, Y}$ is required to evaluate a classifier, and $P_{X, Y}$ remains unspecified even if $P_{X, Z}$ and $P_{Y \mid Z}$ are known. Due to such an underspecification, we are required to derive the Frechet bounds to provide upper and lower bounds for the (underspecified) classifier performance.
>
> **Small labeled set:** Thank you for your comment. We agree that a small labeled sample might be available in some cases, but even for these cases, it can take some time until labels are obtained. Moreover, in many situations, labeling is just too expensive or impossible. It is also true that Snorkel can use a small dev set, but that is **not** needed.
>
> Regarding your suggestion on comparing the confidence interval given by a labeled sample, it suffices to look at the (upper) "tennis" plot in Figure 1. Even though the test set size is large (>1k data points), we can see that the green confidence interval is larger than the distance between the blue and red lines. The green confidence interval would be much bigger if the dataset were smaller, making our approach a great alternative to a small number of labels. Please also check Section A.4, where our new experiment shows that our bounds can be more effective in model selection than small labeled test sets.

---

> ### Author Response · Authors · 2023-11-23
> **Last updates**
>
> Dear reviewer,
>
> In our last paper update, we included in the main text the experiments about model selection using our own method for model selection vs. using a small labeled sample.
>
> Thank you!

---

### Official Review · Reviewer_2tYp · 2023-11-01

**Soundness:** 2 fair
**Presentation:** 3 good
**Contribution:** 2 fair
**Rating:** 5
**Confidence:** 4

**Summary:**

This work develops methods for estimating Fréchet bounds for distributions where some variables are continuous valued and others are discrete. Importantly their method boils down to solving a convex program and does not scale with the dimensionality of the feature space. They derive the asymptotic distribution for their estimators, allowing them to quantify uncertainty. Finally, they cast the main motivating application of their work to be evaluating the performance of models trained under programmatic weak supervision. They compare their bounds to actual test metrics on 5 different PWS datasets, pointing out that they reveal useful insights for PWS trained models.

**Strengths:**

I think the problem setting is well-motivated from a weak supervision point of view. The premise of PWS is that ground-truth labels are scarce (or expensive to procure). Yet often in this field, methods assume access to validation/test labels for model selection/evaluation. Being able to estimate performance *without* requiring (m)any ground-truth labels is important so as to be able to reduce reliance on this assumption and perform more realistic forms of model selection. The approach proposed by this paper is interesting technically and nicely comes with uncertainty estimates.

**Weaknesses:**

The main weakness of the paper in my opinion is in the experiments, which the authors use to show that their bounds have practical usefulness in PWS. However, I think this can be better substantiated in several ways.

For starters the experiments would ideally be expanded to encompass a wider array of PWS settings, i.e.:
- Testing on multiple types of end models (beyond logistic regression) and label models (beyond just Snorkel)
- Testing on datasets beyond binary classification
- Measuring precision/recall on the unbalanced datasets such as *Tennis* and *Census*. On these tasks, F1 (and not accuracy) is the default metric in WRENCH.

But perhaps more importantly, I think the results would benefit from more directly showcasing the usefulness of the bounds as a way for **hyperparameter tuning/model selection.** Personally, I think this is the more interesting use case of the bounds. In the current results, it seems that $L$ and $U$ are not always reliable for precisely estimating performance (i.e., they are often wide or fail to capture true test performance due to the assumption on the label model quality). However, I do think the bounds may still be practically useful in that they can help *relatively* rank different models. This is indeed something that the authors allude to in several places, such as when discussing how they can help pick the threshold on *IMDb* or select LFs on *Youtube*. However, I believe this claim should be tested more extensively, perhaps by defining explicit selection rules and comparing the effectiveness of these rules for selecting *all* hyperparameters (and for more complex WS methods).

Furthermore, I think some relevant baselines for evaluation/model selection here would be (a) using weak labels for the validation set; (b) using a small amount of clean labels. The latter is not a totally fair comparison, but would reflect an important practical question re: "how many labeled points does using these bounds actually save?" Even for the current results, it would be nice to compare the widths of the bounds to something like the 95% CIs one would obtain when using different amounts of labeled data.

**Questions:**

(1) Curious if the authors have thoughts on what other applications their techniques may have besides validating PWS? Does their approach, for instance, generalize to other applications where one can define discrete $Y$ and $Z$?

(2) One core assumption is that *"Any performance metric based on $h$ cannot be estimated without making extra strong assumptions, e.g. $X \perp Y$, or assuming the availability of some labeled samples."* However, I think in practice, it is likely the case that PWS users have access to *some* (albeit quite limited) set of labels. It would be interesting to see if/how the bounds can be sharpened when you allow for some labeled data to be incorporated. Curious if the authors have any thoughts?

---

> ### Author Response · Authors · 2023-11-20
>
> **Inclusion of a new label model, different classifiers, and multinomial classification datasets:** Thank you for your suggestions! We have taken care of this point. Please see the full response to this issue above in the general comments.
>
> **Use of F1 score:** Thanks for pointing out that accuracy might not be suited for some datasets. We have updated the paper using the F1 score when needed.
>
> **Model selection procedure/experiment:** Thank you for suggesting this new idea. We have taken care of this point. Please see the full response to this issue above in the general comments.
>
> **Sharpening bounds using labeled data:** We could possibly include new constraints on the optimization problems we solve to compute $L$ and $U$ based on the observed labeled data. However, that would drastically change our method.
>
> **Applications beyond PWS:** Thank you for your question. Indeed, our method can be applied in different situations. Please check our second paragraph of the "Conclusion" section where we talk about statistical matching.

---

> > ### Comment · Reviewer_2tYp · 2023-11-22
> >
> > Thanks for the detailed response and additional results/explanations! Overall, I like the additions and think they are all steps in the right direction. Before I feel confident enough about updating my score though, I do have a few follow-up questions about new experiments:
> >
> > **Model Selection (Appendix A.4)**
> > * Just to clarify, the models here are trained on labels produced by Snorkel? Was there any selection done on Snorkel's hyperparameters?
> > * Was early stopping also tuned for using your bounds? Typically, this is done for model training in WRENCH (and quite important, at least for the higher performing BERT/RoBERTa models).
> > * Could you also report the final performances of the best model (according to each selection procedure) in addition to the correlation? I think ultimately, the final model performance matters more to practitioners.
> >
> > **Expanded Results (Appendix D)**
> > * The test accuracies for *Trec*, *Semeval*, and *Chemprot* seem quite a bit lower when compared to logistic regression results in Table 11 of the Wrench paper? I suspect this might be because you are using features from `all-MiniLM-L6-v2` instead of `bert-base-cased` but this would be good to get to the bottom of. If this is the explanation, i think in general it would be nice to have results for the more performant (and thus likely more practically relevant) features.
> >
> > Also as a generic comment, I do think given all the new results, there is a non-trivial amount of work still to be done in terms of re-organizing and polishing (e.g. integrating some of the multiclass and model selection in the main). This does make me a bit hesitant about the current form of the manuscript.

---

> ### Author Response · Authors · 2023-11-23
>
> Dear reviewer,
>
> Thank you for your questions. We updated the paper with your requested final model performances and moved all the extra experiments to the main text. Now, the paper is reorganized and polished. Regarding your other questions:
>
> - **Snorkel:** For the experiments in Appendix A.4, we use Snorkel as a label model and then train the classifiers using the noise-aware loss described by [3]. We do not make any hyperparameter selection for Snorkel's model since we assume no access to labeled samples throughout the paper; then we just fit Snorkel with the default configuration.
>
> - **Early stopping:** To train the MLPs in Appendix A.4, we fix the number of epochs but tune the learning rate and regularization parameter, which should have a similar effect.
>
> - **Performance in multinomial classification:** There are two reasons for the performance gap. The most important was that we were only using 10 weak labels (WLs) for each dataset. To use the "Oracle" approach to estimate $P_{Y|Z}$, the number of samples needs to increase exponentially with the number of WLs, making estimation not feasible when the number of WLs is big. To make Snorkel and FlyingSquid results comparable, we also used only 10 WLs. To solve this issue, we now have two sets of results (Tables 5 and 6), whereas, in the second one, we do not subsample WLs. The results are close to Wrench's now, except for SemEval. We believe that this difference is due to the fact that we do not tune our label models and classifiers using any labeled samples, which is done in Wrench's paper (we assume no labeled samples throughout the paper). Tuning the label model in this task is a relevant factor because the number of parameters to be estimated is big (there are many WLs in SemEval). In summary, the results are not comparable.
>
>
> ## References
>
> [3] Ratner, A. J., De Sa, C. M., Wu, S., Selsam, D., & Ré, C. (2016). Data programming: Creating large training sets, quickly. Advances in neural information processing systems, 29.

---

### Official Review · Reviewer_RUBf · 2023-11-04

**Soundness:** 2 fair
**Presentation:** 3 good
**Contribution:** 3 good
**Rating:** 6
**Confidence:** 4

**Summary:**

The paper studies the problem of estimating Frechet bounds. In short, the idea is that we want to evaluate the expectation of a function g(X,Y,Z) with respect to some unknown joint distribution $P_{X,Y,Z}$, but we only have access to some information about this joint distribution, e.g., $P_{Y}$, $P_{Y|Z}$, and $P_{X,Z}$. In Frechet bounds, we compute an upper bound $U$ and lower bound $L$ to the value of this expectation by considering respectively the higher value of the expectation and lower value of this expectation that can be obtained by a joint distribution $P'_{X,Y,Z}$ that satisfies the information that we are given about the marginals.

The paper provides a computationally efficient method to estimate those upper bound $U$ and lower bound $L$ from a set of samples that are used to estimate those marginal informations. Under some assumptions, it also shows asymptotic convergence results of those estimated lower and upper bound to the true values $L$ and $U$ to the their true value when the number of samples goes to infinity.

The estimation of the Frechet bounds is showcased for the problem of evaluating programmatic weak supervision methods. In weak supervision, a classification model $h : X \mapsto Y$ is trained only using the information Z provided by a set of weak supervision sources, since there is no labeled data. While programmatic weak supervision is an important method in practice with a lot of useful applications, it is hard to evaluate the performance of a trained classifier due to the lack of labeled data.

Even without labeled data, the Frechet bounds can be used to upper bound and lower bound  the expected accuracy of a classifier h with the function g(X,Y,Z) = 1(h(X) = Y). The paper shows a practical application of this idea in different benchmark datasets used in weak supervisions. This method can enable the practitioners to evaluate the weak classifiers in a weak supervision setting without access to labeled data.

**Strengths:**

The paper studies a very important problem. Programmatic weak supervision is a method of paramount importance in practice, and the lack of labeled data makes it hard to validate the performance of those models. The paper proposes a general method that can also be used to address this issue.

I think the use of optimistic and pessimistic argument on the missing information (i.e., the missing information of the joint distribution X x Y x Z) to provide bounds is a clever and practical way to get around the lack of information. The paper provides a theoretical discussion of their estimator with respect to computability and asymptotic convergence. I found the application of the smoothing operation to simplify the problem while maintaining approximation guarantees (2.3) very interesting.

While the paper provides some theoretical results on their estimator, I think that the main strength of the paper lies in the empirical evaluation. In particular, I think that the plots of Figure 1 on benchmark weak supervision dataset illustrate the strength of their methods in providing useful information about the accuracy of the weak supervision classifier.

**Weaknesses:**

(1) While I think that the paper is overall well-written, and it is easy to understand its contribution, I think that the presentation is lacking in some parts.

In my opinion, the paper provides no intuition behind the Frechet bounds and what they represent, and what information they are providing in the weak supervision setting.

In my interpretation, we assume to have access to (or an estimate of)  $P(Y|Z)$, $P(Y)$, and $P(X,Z)$. Let's say that as a function $g$, we use the accuracy $g(X,Y,Z)$ = indicator$( h(X) = Y)$. From  the definition, Frechet bounds look at the "worst-case" and "best-case" joint distribution $X \times Y \times Z$ with respect to the value of the expectation of the function g. In this case, the information on the label $Y$ is given by $Z$. From my intuition (possibly wrong), the worst-case (lower-bound) is given by trying to get as much as possible $g(X) = 0$  (we make a mistake) whenever $h(X)$ is different from the most likely value of $Y$ given by  $P(Y|Z)$. Analogous intuition for the upper bound. In particular, it looks like the gap between the lower bound and upper bound depend on "how much" the output of the classifier $h(X)$ differs from $P(Y|Z)$ given a sample $(X,Z)$. If this interpretation is correct, then it looks like that the lower bound and upper bounds are informative (i.e., close to one another) only if $h(X)$ follows the output of the labels $P(Y|Z)$, in which case it is not clear what is the advantage of training a classifier h.

------

(2) I think that the assumption that we can estimate $P_{X,Z}$ is too strong and not necessary (and not really used).

If $X$ is highly dimensional (i.e. images), it is impossible to estimate this distribution within any reasonable finite sample size. I do not think that this assumption is actually necessary for the scope of this paper, in fact all the applications of the results could be formulated substituting $X$ with $h(X)$, in which case $h(X)$ has the same support than the label space Y. In fact, for precision recall etc, we only need to consider a function$ g(Y',Y,Z)$ where $Y'$ is the output of the classifier that is being evaluated.

I find it surprising that the results do not add any assumption on estimating the distribution $(X,Z)$ as this is a very hard problem in general. I think this is due to the fact that we never actually estimate the distribution itself, but we simply evaluate the function g that is bounded. This is a bit misleading with the introduction, where it is written that we use an estimate of $P(X,Z)$.

------

(3) I think that the Appendix section with the proof of the results is not well written nor easy to follow. I believe that the paper would greatly benefit from having a more organized proof structure, so that an interested reader can learn about the techniques used.

As an example, results from other papers are often used (e.g., Beiglbock & Schachermayer for  proof of Theorem 2.1). It takes a lot of effort by the reader to understand the applications of those results, as these cited theorem use some assumptions, have different notation, and it is not immediately straightforward how they are translated to this paper setting. For readability, It would be easier to say something along the lines "since [... these assumptions hold ...] then we can apply this result from [citation] that says: "

Another example is the Proof of Theorem 2.4 (e.g., sequence of equations of page 16, or the sequence of Theorem and Lemmas). It would be easier if the proof is organized in a way such that it is easy to follow and understand the different parts of the proof.

----

(4) Concerns on the experiments.

In the experimental evaluation, only four datasets from Wrench are used. Wrench provides a lot of different datasets, it is not clear why only 4 were selected for the evaluation, and why only those specific 4 were selected (there are other datasets that are also limited to binary classification as SMS, Commercial ... from Wrench). In some practical cases in Wrench (as Youtube), we also care about the F1 score, which I think it can also be estimated with your method.

----

(5) Assumption on being able to estimate $ P(Y|Z) $.

The assumption that one can estimate P(Y|Z) is very strong, and this is how the paper gets around the need of labeled data. This is unrealistic in practice. I think that the paper should specify that happens when the model is misspecified (theory), which is often the case for the parametric model mentioned in the paper. It would be interesting to understand if we can say something about the Frechet bound if we allow for a bounded misspecification, i.e. $|P(Y|Z) - \hat{P}(Y|Z)| < Delta$ for some $Delta >0$.

---

There are other papers that used somehow similar ideas to consider a "worst-case"(or adversarial) distribution over all possible distributions in a weak supervision setting that I think are very relevant (e.g.,  [A,B,C,D,E,F]). In particular, this problem was studied mostly  for the case of trying to solve the map Z -> Y from weak supervision sources outputs to label without adding any assumption (e.g., independence or parametric assumption) [A,B,E], although some other work consider the more general case of training a classifier h [C,D,F].

The use of the "worst-case" allows to obtain an upper bound (analogous to the U of the Frechet Bound). As an example, the underlying idea is to consider possible joint distribution $Z,Y$ that satisfy some information on the marginal $P(Y|z_i)$ or $P(Z)$ (this information may vary among the work).

[A]: Balsubramani, A. and Freund, Y. Optimally combining classifiers using unlabeled data. In Conference on Learning Theory (COLT), pp. 211–225, 2015.
[B]: Balsubramani, A. and Freund, Y. Optimal binary classifier aggregation for general losses. In Neural Information Processing Systems (NeurIPS), 2016.
[C]: Arachie, C. and Huang, B. Adversarial label learning. In AAAI Conference on Artificial Intelligence (AAAI), 2019.
[D]: Arachie, C. and Huang, B. A general framework for adversarial label learning. The Journal of Machine Learning Research, 22:1–33, 2021.
[E]: Mazzetto, A., Sam, D., Park, A., Upfal, E., and Bach, S. H. Semi-supervised aggregation of dependent weak supervision sources with performance guarantees. In Artificial Intelligence and Statistics (AISTATS), 2021.
[F]: Mazzetto, A., Cousins, C., Sam, D., Bach, S. H., and Upfal, E. Adversarial multi class learning under weak supervision with performance guarantees. In International Conference on Machine Learning, pp. 7534–7543. PMLR, 2021a.

Note that some of this work may require to have some labeled data to get this marginal information. Some information about the label is required in general. In some work, they assume to have access to some labeled data to estimate some statistical constraints. On the other hand, the result of this paper removes this assumption,  but assume that you can get information of the labels from the label model P(Y|Z) (however this model could be misspecified).

**Questions:**

I think the paper studies an important problem and provides an interesting solution to it, and I am open to raise the score if the following concerns are addressed (summarized below, see Weakness for additional details).

1. What is the interpretation of the Frechet bounds? When is the lower bound close to the upper bound, and under which conditions are those bounds informative.

2. What happens to your results when $P(Y|Z)$ is misspecified from a theory point of view?

3. Why did you use only 4 binary classification tasks from Wrench, and why those specific four were selected? Are the results different for the other tasks?

---

> ### Author Response · Authors · 2023-11-20
>
> **Interpretation of the Fréchet bounds:** A more direct intuition is seeing Fréchet bounds using ideas from optimal transport (more details in Section A.1). In summary, the Fréchet bounds find the worst and best cases for the distributions $(X, Y) \mid Z = z$ with respect to the expected value of $g$ for each value of $z$. More explicitly speaking, for each $z$, the objective for the Fréchet bound finds the maximum and minimum values for $E_{P_{X, Y\mid Z = z}}[g(X, Y, z)]$ when $P_{X \mid Z = z}$ and $P_{Y \mid Z = z}$ are specified. That is, for each $z$, we solve an optimal transport problem. Consequently, if $g(x,y,z)=\mathbf{1}[h(x)=y]$ and for a fixed $z$, we transport the mass from $P_{X \mid Z = z}$ to $P_{Y \mid Z = z}$ maximizing (resp. minimizing) classification errors in order to compute the lower (resp. upper) bound for accuracy.
>
> Furthermore, we have added a section (Section A.2 in our updated draft) that quantifies the gap between the upper and lower Frechet bounds, where we establish
> $ (U - L)^2 = O(\min \\{ H(X \mid Z), H(Y \mid Z) \\})$, where $H(\cdot \mid Z)$ is the conditional entropy given $Z$. Please see the discussion above in the general comments.
>
> We will include some of this discussion in the main text once the paper is fully reorganized.
>
> **On the estimation of $P\_{X,Z}$:** We appreciate your insight that in practice we care only about $h(X)$ and not $X$ in most cases. However, our method does not require estimating $P_{X,Z}$ in the sense of density estimation. In our Eq. (2.5), we only use samples $\{(X\_i,Z\_i)\}$ to approximate $P\_{X,Z}$ using the empirical distribution $\frac{1}{n}\sum\_{i=1}^n \delta\_{\{(X_i,Z_i)\}}$ ($\delta$ is the Dirac measure) and, therefore, approximating expectations using empirical means.
>
> **Reorganizing the proofs:** Thank you very much for your observation. We will reorganize the proofs in the next version of the paper to make them easier to understand.
>
> **Expansion of experiments and inclusion of F1 score:** Thank you for your suggestions. Please check the general comments above for a detailed explanation of what we have added.
>
> **Label model misspecification:** Thank you for your suggestion on this point. Please find the response to this issue in the general comments above.
>
>
> **Related work:** Thank you for your suggestions for new references. Adversarial learning is indeed related to our work in some ways (but not a substitute). We will update the "Related work" section and incorporate your suggested references in the final version of this paper.

---

> ### Author Response · Authors · 2023-11-23
> **Last updates**
>
> Dear reviewer,
>
> In our last paper update, we included, in the main text, the adversarial learning references (Related work) and a discussion about the new theoretical results (Section 2.4).
>
> Thank you!

---

> > ### Comment · Reviewer_RUBf · 2023-12-04
> > **Response**
> >
> > I would like to thank the authors for their detailed responses. I appreciate the effort of the authors to improve the paper, and I believe the interpretation of the Fre'chet bounds makes the paper clearer!
> >
> > After reading the other responses and the answers of the authors, I would like to keep my score (I would give 6.5 if I could), leaning towards acceptance.

---

### Author Response · Authors · 2023-11-20
**General response to all reviewers**

## Thank you!

We thank all reviewers for their constructive and insightful comments. This section provides general responses to the common concerns shared among multiple reviewers and comments about the main increments in the paper. Please let us know if you have further questions!

## New theoretical results

**Bounds for F1 score:** We have included in Section 3.1.2 estimators and results for the F1 score.

**Interpreting the Fréchet bounds through the lens of optimal transport:** We have included in Section A.1 of the appendix a discussion on how to interpret the Fréchet bounds through the lens of optimal transport. This can help readers to intuitively understand what is behind the scenes when Fréchet bounds are computed. We will include some of this discussion in the main text once the paper is fully reorganized.

**Label model misspecification:** We have included in Section A.3 of the appendix a new theoretical result regarding the label model misspecification. In that result, we assume that the label model $\hat{P}\_{Y|Z}$ converges to a certain distribution $Q\_{Y|Z}$ such that $d_{\text{TV}}(P\_{Y|Z=z},Q\_{Y|Z=z})<\delta$ for all $z$, and then we prove that the resulting bounds will be at most a distance of $C\delta$ from the true bounds, for a certain positive universal constant $C$. We recognize that if $\delta$ is large there is no guarantee that our bounds will be meaningful. However, there is also no guarantee that programmatic weak supervision works at all in that case. The good news is that the computed bounds seem to be very robust to the model specification from our experiments (most of the time, the bounds contain the test performance). We will include some of this discussion in the main text once the paper is fully reorganized.

**When are the Fréchet bounds informative?** We have included in Section A.2 of the appendix a new theoretical result regarding the size of the difference $U-L$ (upper bound minus lower bound). If this difference is small, we can accurately estimate $\mathbb{E}[g(X,Y,Z)]$ without observing samples from the joint distribution of $(X,Y,Z)$. Intuitively, the new theorem says that if the uncertainty we have about $X$ or $Y$ after observing $Z$ is low, then the bounds $L$ and $U$ should be close (we bound that difference using the concept of conditional entropy). For example, take the extreme case where $Y=Z$: knowing $Z$ tells us exactly the value of $Y$, and we could perfectly label every data point $(X,Z)$, recovering the triplet $(X,Y,Z)$. In that extreme case, the conditional entropy of $Y$ given $Z$ is zero, implying $U=L$.



## Expansion of experiments

**New Wrench datasets and use of F1:** We have included in our paper all Wrench classification datasets (according to Table 1 of [1]). Not all results are displayed in the main text due to space constraints, and some were moved to Appendix D. Generally, results are promising, with the bounds containing the test scores. We now use the F1 score (instead of accuracy) in the same datasets as [1].

**Inclusion of a new label model, different classifiers, and multinomial classification datasets:** We included FlyingSquid as a new label model (please check Appendix D), and neural networks in our experiments (please check Appendix A.4.2). Moreover, we included multinomial classification datasets from Wrench, i.e., AGNews, TREC, SemEval, ChemProt (please check Appendix A.4.2 and D).

**Model selection procedure/experiment:** We describe in Section A.4 of the appendix some possible ideas on how to use our method to make model selection. Moreover, we ran an extra experiment showing that our bounds are very useful in ranking models even when compared to the situation in which a small labeled sample is available. Another baseline we consider is using the weak labels to rank classifiers (through the label model), which most of the time showed to be an inferior approach. We will include some of this discussion in the main text once the paper is fully reorganized.


## References

[1] Zhang, J., Yu, Y., Li, Y., Wang, Y., Yang, Y., Yang, M., & Ratner, A. (2021). Wrench: A comprehensive benchmark for weak supervision. arXiv preprint arXiv:2109.11377.

---

### Author Response · Authors · 2023-11-23
**Last updates: general response to all reviewers**

Dear reviewers,

I have made the final changes to our paper. Namely, we included:
- References suggested by reviewers RUBf and JyZF in the "Related work" section;
- A discussion about the new theoretical results in Section 2.4;
- New experiments in the main text (they were previously in the appendix since our last update).

Thank you!

---

### Meta-Review · Area_Chair_fRXU · 2023-12-05

**Metareview:**

This is a borderline paper with both arguments for accepting and rejecting the paper. After some deliberation I recommend to reject the paper because there is a significant gap between when the theoretical analysis of the paper is valid and when weak labeler set selection is well motivated. I believe this should be addressed in a major revision before publication.

**Justification For Why Not Higher Score:**

n/a

**Justification For Why Not Lower Score:**

n/a

---

### Decision · Program_Chairs · 2024-01-16

Reject